

# ALPs and HNLs at LHC and muon colliders: Uncovering new couplings and signals

**Marta Burgos Marcos**[⋆]**, Arturo de Giorgi**[†]**, Luca Merlo**[‡] **and Jean-Loup Tastet**[°]

Departamento de Física Teórica and Instituto de Física Teórica UAM/CSIC,
Universidad Autónoma de Madrid, Cantoblanco, 28049, Madrid, Spain

⋆ marta.burgosmarcos@estudiante.uam.es , † arturo.degiorgi@uam.es ,
‡ luca.merlo@uam.es , ° jean-loup.tastet@uam.es

## Abstract

Axion-like particles (ALPs) and heavy neutral leptons (HNLs) are two well-motivated classes of particles beyond the Standard Model. It is intriguing to explore the new detection opportunities that may arise if both particle types coexist. Part of the authors already investigated this scenario in a previous publication, within a simplified model containing an ALP and a single HNL, identifying particularly promising processes that could be searched for at the LHC. In this paper, we first consider the same setup with a broader range of both production processes and final states, both at the High-Luminosity LHC and at a future muon collider. Subsequently, we expand it to the more realistic scenario with at least two HNLs, necessary to describe the active neutrino masses. Different phenomenological signals are expected and we examine the complexities that emerge in this setup. This study paves the way for dedicated analysis at (forthcoming) colliders, potentially pinpointing the dynamics of ALPs and HNLs.

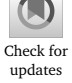

# 1   Introduction

The 2012 Higgs discovery at LHC [1, 2] confirmed the Standard Model (SM) success in describing the strong and electroweak interactions. Since then, no evidence of New Physics (NP) emerged, either in direct searches at colliders or in indirect searches at low-energies facilities. Despite this lack of experimental breakthroughs, significant efforts have been made to look for the most promising NP candidates, necessary to solve various open problems of the SM. In particular, we can mention the search for heavy neutral leptons (HNLs) and axions or axion-like particles (ALPs).

The HNLs are composite fermions originated by the mixing between the neutral components of the SM lepton doublets and exotic sterile leptons, uncharged under any gauge symmetry of the SM and usually described as right-handed (RH) neutrinos, which play a role in the active neutrino mass generation mechanism. The active neutrinos are the resulting light mass eigenstates, while the HNLs are the heavy ones and their masses can span various orders of magnitude, depending on the specific mechanism and naturalness hypotheses of the Dirac Yukawa couplings. The best-known proposal is the Type-I Seesaw mechanism [3–6], where the smallness of the active neutrino masses is due to the heaviness of the HNLs. In the original version, all leptons are charged under Lepton number (LN) in the same way and the Majorana mass term of the sterile neutrinos represents an explicit breaking of LN. Moreover, this term is not protected by any symmetry and the Majorana mass can be taken arbitrarily large. It follows that the active neutrino mass scale can be correctly reproduced with HNL masses of $\mathcal{O}(10^{14})$ GeV and Dirac Yukawas naturally of $\mathcal{O}(1)$. In the generic Type-I Seesaw realisation, the Majorana mass is the scale that suppresses any contribution at low-energy of the HNLs, that is the active neutrino masses described by the Weinberg operator [7] and any other non-renormalisable operator obtained integrating out the HNLs. With such a large Majorana mass, apart from the active neutrino oscillations, these effects can hardly be observed experimentally [8].

It is worth noting the possibility that, within the same context, the Dirac Yukawa terms might be tiny, perhaps on the order of the electron Dirac Yukawa, approximately $10^{-6}$. Consequently, the HNL Majorana mass scale could be as low as around 100 GeV. One might then expect to detect the HNLs directly at colliders. However, their interactions are suppressed by the mixing between the lepton doublets and the sterile leptons that linearly scales with the Dirac Yukawas. As a result, even in this scenario, the effects of HNLs at low energies are unlikely to be noticeable at current and future colliders.

Alternative constructions where the HNLs are relatively light — let's say $\sim$ TeV scale — with possibly visible effects while still reproducing correctly the active neutrino masses, go by the name of Low-Scale Seesaw mechanisms [9–12]. The lightness of the active neutrino masses is not due to the heaviness of the HNLs but to specific structures of the Dirac and Majorana mass terms, determined by certain LN assignments of the sterile neutrinos. The peculiar aspect of these constructions is that the LN is almost an exact global symmetry of the Lagrangian.

There exists a vast experimental program searching for HNLs over a wide range of masses, in both the prompt and displaced regimes (see e.g. Refs. [8, 13, 14] for detailed reviews of existing and proposed searches). At the LHC such searches include CMS [15–21], ATLAS [22–25] and LHCb [26]. HNLs are also a prime target for current and future searches at displaced detectors [27–38], extracted beamlines [39–56], colliders [57–72], and cLFV experiments [73, 74].

On the other side, Axions represent the best-motivated candidates to solve the Strong CP problem. In the traditional models [75–81], the QCD axion arises as a Goldstone Boson (GB) of an Abelian global symmetry anomalous with QCD, the so-called Peccei-Quinn (PQ) symmetry. By construction, its mass is protected and originated by the explicit breaking of the $U(1)_{PQ}$ due to non-perturbative QCD dynamics. The characteristic aspect of the QCD axion is the inverse proportionality between its mass and its scale $f_a$, which leads to the popular QCD axion band in its couplings parameter space. Very much recently, a series of works proved the viability of axions that solve the Strong CP problem while living outside the QCD axion band [82–90], sparking interest of axion searches in broader regions of the parameter space.

The word ALP refers to any particle that resembles the QCD axion, but do not necessarily solve the Strong CP problem. In particular, its mass and its characteristic scale $f_a$ are independent parameters. They have been studied in various contexts, such as in flavour dynamics [91–100], in composite Higgs frameworks [101–103], in association to the breaking of the Lepton numbers [104–113], in cosmology [114–119], including the possibility to solve the Hubble tension [95, 120–122]. Alongside with this more theoretical effort, the search for ALPs at experiments is extremely active. We can mention studies both at low-energy facilities [97, 123–137] and at colliders [138–159].

A generic feature of axions and ALPs is that their shift-symmetry invariant couplings to fermions are proportional to the fermion masses. This motivated the analysis in Ref. [160], where the authors identified a particularly clean process at the LHC that arises from the co-existence of a single ALP and a single TeV-scale HNL. More into detail, this ALP is produced through gluon fusion and decays into two on-shell HNLs, both of which subsequently decay into a charged lepton and an on-shell $W$ gauge boson. The latter decays into jets and the resulting final state consists of four jets (4j) and two charged leptons ($2\ell$) with the same or opposite signs. Such topology was termed by the authors JALZ to recall the final state particles ($j4\ell2$) (a similar one has been studied in Ref. [161] in the context of a $Z'$-portal to HNLs and in Ref. [162] for the type-III Seesaw). The advantages are multiple: in this process, the ALP-HNL coupling is proportional to the HNL mass, thus enhancing the signal strength; as the HNLs are on-shell, it is possible to adopt the narrow-width approximation and the dependence on the small mixing angle between the active and the RH neutrinos fades away. Moreover, there is no SM background when the two leptons have the same charge (breaking lepton number) or different flavours (breaking lepton flavour); the final state is fully reconstructible, with a highly-specific kinematic structure with four simultaneously on-shell particles. The background efficiency will crucially depend on the ability of the detector to resolve the HNL mass, and is generically expected to scale with the square of the resolution. For the remaining processes with opposite-sign and same-flavour final state leptons, a dedicated analysis is likely required to precisely estimate the SM background (as well as other sources of background, such as e.g. combinatorial background), that should be suppressed due to the specific kinematic structure of the considered final state.

The conclusion of Ref. [160] is that the study of the interplay between ALPs and HNLs may shed light on the nature of these elusive particles and allows to extract joint bounds that are much stronger than those on the individual particles taken separately. We extend the study in Ref. [160] to a broad range of possible production mechanisms of the ALP as well as to various final states. Besides considering the present and future phases of the LHC, we discuss the case

of a proposed 10-TeV muon collider [163]. Moreover, we discuss a more realistic scenario with two HNLs, which is the minimum number required to correctly describe the active neutrino masses and the PMNS mixing matrix. This is not a simple extension of the single HNL case, as the heaviest HNL can decay into the lightest one while simultaneously emitting an ALP — which we dubbed the cascade case — giving rise to a different final state with respect to the JALZ one. Also, in this more general setup, the joint bounds that we extract when the ALP and HNLs coexist are stronger than those available in the literature when considering these NP particles separately. Furthermore, the pure JALZ events and the cascade processes are sensitive to different couplings of the ALP to the HNLs, thus representing a unique possibility to investigate the flavour structure of these couplings.

The interplay between ALPs and HNLs is nowadays a hot topic and this work complements other analyses already appeared in the literature: studies at beam-dump experiments for very light long-lived particles [164, 165]; Leptogenesis via ALP-HNL couplings for very heavy ALPs [166]; and consequences of an ALP portal to HNLs as DM candidates [167].

The structure of the paper can be read in the Table of Contents.

## 2 The lagrangian description

This section is devoted to illustrating the formal description of HNLs and ALPs. We first present an overview of the Seesaw mechanisms that include HNLs to provide model-building support to the numerical analysis in the next sections. Subsequently, we define the effective Lagrangian we will adopt for the rest of the paper including an ALP and HNLs.

### 2.1 Seesaw mechanisms with HNLs

Considering the purely leptonic sector, the generic Type-I Seesaw Lagrangian is given by

$$\mathscr{L}^{\text{Type-I}} = i\,\overline{N_R'}\,\slashed{\partial}\,N_R' - \left( \overline{L_L'}\,\widetilde{H}^\dagger\,Y_D\,N_R' + \frac{1}{2}\overline{N_R'^c}\,\Lambda\,N_R' + \text{h.c.} \right), \tag{1}$$

where $N_R'$ are the RH neutrinos, singlets of the full SM gauge group, $N_R'^c \equiv \mathcal{C}\overline{N_R'}^T$ with $\mathcal{C}$ is the charge conjugation matrix, and the prime refers to the flavour basis. The RH neutrinos only interact with the SM particle content through the Dirac mass term with active neutrinos, proportional to $Y_D$. The popular description in terms of LN is with all the leptons with the same charge assignment, $L(L_L') = L(N_R')$, such that the whole Lagrangian is LN preserving at tree-level, with the exception of the Majorana mass term, proportional to $\Lambda$, that violates LN by two units.

After the Electroweak Spontaneous Symmetry Breaking (EWSB), the following mass term for the neutral fields is generated: indicating the neutral leptons within a unique vector $\chi_L' \equiv (\nu_L', N_R'^c)$,

$$\mathscr{L}_\chi \supset -\frac{1}{2}\overline{\chi_L}'\,\mathcal{M}_\chi\,\chi_L'^c, \qquad \text{with} \qquad \mathcal{M}_\chi = \begin{pmatrix} 0 & m_D \\ m_D^T & \Lambda \end{pmatrix}, \tag{2}$$

where $m_D = Y_D\,v/\sqrt{2}$ represents the Dirac mass matrix with $Y_D$ a $n \times 3$ matrix containing the Yukawa couplings and $n$ the number of HNL species introduced. On the other hand, $\Lambda$ is an $n \times n$ matrix responsible for Majorana-like mass terms for RH neutrinos.

To move to the mass basis, we can start performing a block-diagonalisation of the mass matrix given in Eq. (2). By doing so, a mixing $\Theta$ between SM and RH neutrinos is generated,

relating the original flavour eigenstates $\chi'_L$ and the block-diagonalised eigenstates: assuming that the entries of $\Theta$ are small, at first order in the expansion we can write

$$\nu'_L \to \nu'_L + \Theta N'^c_R, \qquad \text{with} \qquad \Theta = m_D \Lambda^{-1}. \tag{3}$$

The resulting mass matrix for the light active neutrinos is given by the traditional Type-I Seesaw expression, that in the limit $||\Lambda|| \gg v$ reads

$$m_\nu \simeq -\Theta \Lambda \Theta^T = -m_D \Lambda^{-1} m_D^T, \tag{4}$$

while the HNLs have a mass matrix that approximately coincides with $\Lambda$. The mass basis is finally obtained diagonalising $m_\nu$ and $\Lambda$ with unitary transformations. In particular, the diagonal active neutrino mass matrix can be written as

$$\widehat{m}_\nu = U_L^\dagger m_\nu U_L^c, \tag{5}$$

where $U_L$ represents the PMNS mixing matrix, performing this last rotation in the mass basis for the charged leptons.

The active neutrino masses are reproduced with a Majorana mass $\Lambda \sim \mathcal{O}(10^{14})$ GeV and Dirac Yukawas $Y_D \sim \mathcal{O}(1)$, that we consider as a natural value according to the t'Hooft naturalness principle. Alternatively, we may assume that the Dirac Yukawas are unnaturally suppressed, or an underlying mechanism is in action to suppress them, and as a result, the Majorana mass can be much smaller: for example, if we take the same scale as the electron Yukawa $Y_D \sim 10^{-6}$, then $\Lambda \sim 10^2$ GeV. In both cases, the mixing $\Theta$, whose presence implies that the HNLs acquire couplings with the EW gauge bosons, is very much suppressed, $\Theta \sim 10^{-13}$ in the first case and $\Theta \sim 10^{-7}$ in the second. It follows that the HNLs will hardly be produced and detected at colliders. On the other hand, it is possible to integrate them out leading to a low-energy non-realisable operator. However, it turns out to be proportional to the combination $\Theta\Theta^\dagger$ that, given its smallness, makes the corresponding contributions practically invisible. All in all the Type-I Seesaw mechanism, although very elegant and simple, is hardly testable and the main reason is that the smallness of the active neutrino masses is due to the largeness of the Majorana mass scale that also suppresses the $\Theta$ mixing.

It is possible to decorrelate the smallness of the active neutrino masses and the suppression of the low-energy HNL effects in versions of the Type-I Seesaw mechanism that are known as Low-Scale Seesaw (LSSS) constructions [9–12]. The main feature is that the LN is not strongly broken by the Majorana mass term, but instead, it is an almost exact symmetry of the Lagrangian. For this reason, this class of constructions is also known as "LN protected" Seesaw mechanisms. The HNLs can be relatively light in these scenarios and therefore they could be produced at colliders and their low-energy effects are expected to be visible in indirect searches. Moreover, given the smallness of the explicit LN breaking, the HNLs turn out to be pseudo-Dirac pairs, with almost degenerate masses (the splitting is proportional to the active neutrino masses), having interesting phenomenological implications in the low-energy observables, such as the suppression of LN breaking effects [10].

Having a Majorana mass term that preserves LN implies that the RH neutrinos transform differently under LN, opening up various possible realisations of the LSSS models. It is very common in the literature to prefer a slightly different notation in these types of constructions, where the RH neutrinos are grouped into (at least) two classes: we will call them $N'_R$ and $S'_R$, which only differ in the LN charge assignment.

A popular choice is $L(L'_L) = L(N'_R) = -L(S'_R)$ which leads to the following LN conserving Lagrangian:

$$-\mathscr{L}_{\text{LN}} = \overline{L'_L} \, \widetilde{H} \, Y_N \, N_R + \frac{1}{2} \left( \overline{N'^c_R} \, \Lambda \, S'_R + \overline{S'^c_R} \, \Lambda^T \, N'_R \right) + \text{h.c.}, \tag{6}$$

where $Y_N$ is a Dirac Yukawa matrix of $3 \times n$ dimension, and $\Lambda$ is a $n \times m$ matrix, being $n$ ($m$) the number of $N'_R$ ($S'_R$) RH neutrino fields. Active neutrino masses can only be described by introducing (a combination of) additional terms, explicitly violating the LN symmetry: in all generality, we can write

$$-\mathscr{L}_{\epsilon\text{LN}} = \overline{L'_L}\,\widetilde{H}\,\epsilon\,Y_S\,S'_R + \frac{\mu'}{2}\,\overline{N'^c_R}\,N'_R + \frac{\mu}{2}\,\overline{S'^c_R}\,S'_R + \text{h.c.}\,, \tag{7}$$

where $Y_S$ is a Dirac Yukawa matrix of $3 \times m$ dimension, and $\mu$ ($\mu'$) is a $m \times m$ ($n \times n$) matrix. According to the t'Hooft naturalness principle, the norms of the three quantities $\epsilon Y_S$, $\mu$ and $\mu'$ should be small compared to the LN preserving ones appearing in Eq. (6).

After the EWSB, we can write again Eq. (2), making explicit the entries of the RH neutrino sector: taking now $\chi'_L \equiv (\nu'_L, N'^c_R, S'^c_R)$, we have

$$\mathcal{M}^{\text{LSSS}}_\chi = \begin{pmatrix} 0 & m_N & \epsilon\,m_S \\ m_N^T & \mu' & \Lambda \\ \epsilon\,m_S^T & \Lambda^T & \mu \end{pmatrix}. \tag{8}$$

At leading order in the $\mu^{(\prime)}/\Lambda$ and $\epsilon m_S/\Lambda$ expansion, the active neutrinos mass matrix is given by

$$m^{\text{LSSS}}_\nu \simeq -m_N \frac{1}{\Lambda^T} \mu \frac{1}{\Lambda} m_N^T - \epsilon\left(m_S \frac{1}{\Lambda} m_N^T + m_N \frac{1}{\Lambda^T} m_S^T\right). \tag{9}$$

With this result at hand, we can easily see that the HNLs do not need to be extremely heavy to reproduce the smallness of the active neutrino masses: for a chosen $\Lambda \sim \mathcal{O}(\text{TeV})$, it is sufficient to fix the norms of $\epsilon m_S(\mu) \sim 10(1000)$ eV. Notice that $\mu'$ does not appear in the expression above, as the tree-level contribution is strongly suppressed, as it appears suppressed by $\epsilon^2$.

Very interestingly, integrating out the HNLs, the Wilson coefficient of the $d = 6$ operator generated at low energy only depends on the LN violating parameter at the sub-leading level, while the dominant contribution goes with $m_N \Lambda^{-1} \Lambda^{-1\dagger} m_N^\dagger$ that is much larger than the corresponding one in the Type-I Seesaw scenario. If follows that LSSS constructions describe possibly interesting phenomenological effects in both direct and indirect searches [11].

Two very popular LSSS models are the ones obtained by setting $\epsilon = \mu' = 0$ that takes the name of Inverse Seesaw (ISS) [168, 169], or by imposing $\mu = \mu' = 0$ that is dubbed as Linear Seesaw (LSS) [170, 171]. The expressions for the active neutrino mass matrix in these cases follow Eq. (9) by simply switching off the corresponding LN-violating parameters. It is interesting to underline a difference between the ISS and LSS scenarios. In the first one, it is not possible to successfully describe the neutrino spectrum and the PMNS mixing matrix with only two RH neutrinos, $n = m = 1$, as $\mu$ and $\Lambda$ are just numbers and the product $m_N m_N^T$ has rank one. On the other side, in the second case, the active neutrino mass matrix has rank 2 without enlarging the RH neutrino spectrum beyond $n = m = 1$ and allows for a description of the neutrino sector compatible with data as discussed in Ref. [12].

A special discussion is necessary when $\mu'$ is not negligible with respect to the scale of $\Lambda$. This case is dubbed in the literature as Extended Seesaw (ESS) [172, 173] and its main difference with respect to the ISS and LSS resides in the relevance of the 1-loop contributions to the active neutrino masses [173]. Indeed, $\mu'$ now represents an explicit large LN breaking and the corresponding contributions, although at the quantum level, are as important as the tree-level ones. Moreover, the presence of a large $\mu'$ breaks the HNLs degeneracy. For simplicity, but without loss of generality, we will restrict to the simple case with only two RH neutrinos, $n = m = 1$, such that $\Lambda$, $\mu$ and $\mu'$ are simple mass scales, while $Y_N$ and $Y_S$ are tridimensional

vectors. The HNLs masses read in this case

$$M_{N_{1,2}} \approx \frac{\Lambda}{2} \left[ \sqrt{4 + \left(\frac{\mu'}{\Lambda}\right)^2} \mp \left(\frac{\mu'}{\Lambda}\right) \right] \approx \Lambda \mp \frac{\mu'}{2},$$  (10)

where in the last step we approximated $\mu' \lesssim \Lambda$ at LO, yielding the heavy neutrinos mass splitting

$$\Delta M_N \approx \mu'.$$  (11)

On the other hand, the leading 1-loop correction to the active neutrino mass matrix in the limit $\nu \lesssim \mu' \lesssim \Lambda$ reads [113, 172–175]

$$\delta m_\nu^{1\mathrm{L}} \approx 2 \frac{m_N m_N^T}{(4\pi \nu)^2} \frac{M_H^2 + 3M_Z^2}{M_{N_1} + M_{N_2}} \log\left(\frac{M_{N_1}}{M_{N_2}}\right) \approx -\frac{Y_N Y_N^T}{32\pi^2} \times \frac{\mu'}{\Lambda^2} \times (M_H^2 + 3M_Z^2),$$  (12)

keeping only the LO contribution. A back-of-the-envelope calculation easily reveals that for a HNL mass scale $M_{N_{1,2}} \sim 1$ TeV and a splitting that varies between 10% and 90%, it is necessary to take $Y_N \sim \mathcal{O}(1) \times 10^{-5}$. In summary, the ESS construction allows for non-degenerate HNLs, although the correct description of the active neutrino masses requires a suppression of the LN-conserving Dirac Yukawa.

## 2.2 ALP-HNL effective lagrangian

We define now the effective Lagrangian that we will adopt in the rest of the paper. We will assume the presence of only one ALP and of only two RH neutrinos in total, which implies $n = 2$ in the traditional Type-I Seesaw and $n = 1$ and $m = 1$ in LSSS ones. This helps simplify the analytical description, but it does not limit the validity of our results that could be easily generalised to higher numbers of RH neutrinos, once focusing on the phenomenology of the two lightest HNLs. In particular, we will not refer to any specific Seesaw mechanism but deal with two HNLs, $N_1$ and $N_2$, whose masses are taken at the TeV scale and labelled as $M_{N_1}$ and $M_{N_2}$.

From the seminal paper in Ref. [176], a big effort has been put to construct a consistent effective description of ALPs [143, 177–183]. Our starting point is a SM gauge symmetry invariant Lagrangian that reads

$$\mathscr{L}_a = \frac{1}{2} \partial_\mu a \, \partial^\mu a - \frac{1}{2} m_a^2 a^2 + \mathscr{L}_a^X + \mathscr{L}_{\partial a}^\psi,$$  (13)

where the first two terms are the ALP kinetic and mass terms, while the last two pieces are the derivative interactions of the ALP with fermions and the ALP anomalous couplings with gauge bosons, respectively.

The derivative shift-invariant ALP couplings to fermions are parameterised by generic $3 \times 3$ Hermitian matrix, which encodes their flavour structure. In this work, however, we will assume flavour universal and CP-conserving couplings for each fermion species, except for the RH neutrinos that we allow to be non-universal:

$$\begin{aligned}
\mathscr{L}_{\partial a}^\psi &= \frac{\partial_\mu a}{f_a} \Big[ c_Q \overline{Q}'_L \gamma^\mu Q'_L + c_u \overline{u}'_R \gamma^\mu u'_R + c_d \overline{d}'_R \gamma^\mu d'_R + c_L \overline{L}'_L \gamma^\mu L'_L + c_e \overline{e}'_R \gamma^\mu e'_R + c_N \overline{N}'_R \gamma^\mu N'_R \Big] \\
&= \frac{\partial_\mu a}{2 f_a} \Big[ (c_u - c_Q) \overline{u}' \gamma^\mu \gamma_5 u' + (c_d - c_Q) \overline{d}' \gamma^\mu \gamma_5 d' + (c_e - c_L) \overline{e}' \gamma^\mu \gamma_5 e' \\
&\qquad + 2 c_L \overline{\nu}'_L \gamma^\mu \nu'_L + 2 c_N \overline{N}'_R \gamma^\mu N'_R \Big],
\end{aligned}$$  (14)

where $Q_L'$, $u_R'$ and $d_R'$ are the SM quark doublet and singlets, while $L_L'$ and $e_R'$ are the SM lepton doublet and singlet. With $N_R'$ we refer to the RH neutrinos that may have equal or different LN charges, depending on the specific Seesaw mechanism considered. In the second line, we made explicit the couplings below EWSB. The flavour universality condition implies that $c_Q$, $c_u$, $c_d$, $c_L$ and $c_e$ are just dimensionless numbers, while $c_N$ is an Hermitian $2 \times 2$ matrix. This *ad hoc* condition has in general a deep impact on the associated phenomenology and the effects of non-universal couplings with charged fermions have already been studied in the literature (see Ref. [127] for a recent review). On the other side, relaxing the universality condition on the charged fermion coupling has no impact on our analysis, as will be shown in the next section.

According to the above discussion, the couplings with the charged fermions are unaltered when moving to the fermion mass basis, while the neutral states require a proper discussion. If $c_N \propto \mathbb{1}$, that is universal couplings as for the other fermions, then $\nu'$ and $N'$ mix when moving to the mass basis, but without affecting the couplings with the ALP. As a result, $c_L$ and $c_N$ are already the ALP couplings with the active neutrino mass eigenstates and the HNL ones: by using the fermion equations of motions, these couplings turn out to be proportional to the active neutrino masses and the HNLs masses, respectively. The corresponding phenomenology has been recently studied in Ref. [157] for the ALP couplings with the active neutrinos and in Ref. [160] for the HNLs. On the other hand, if $c_N \not\propto \mathbb{1}$, moving to the mass eigenbasis, the ALP develops a triple coupling with the two different HNL mass eigenstates: this is a new feature concerning Ref. [160] that is responsible for cascade events with a richer associated phenomenology.

On the other hand, the anomalous shift-breaking interactions with gauge bosons are encoded in $\mathscr{L}_a^X$ and can be written as

$$\mathscr{L}_a^X = -\frac{1}{4} c_{\widetilde{B}} \frac{a}{f_a} B_{\mu\nu} \widetilde{B}^{\mu\nu} - \frac{1}{4} c_{\widetilde{W}} \frac{a}{f_a} W_{\mu\nu}^i \widetilde{W}^{i\mu\nu} - \frac{1}{4} c_{\widetilde{G}} \frac{a}{f_a} G_{\mu\nu}^a \widetilde{G}^{a\mu\nu}, \tag{15}$$

where $X_{\mu\nu}$ are the gauge field strengths of the SM gauge bosons and $\widetilde{X}_{\mu\nu}$ their dual. After EWSB, the above Lagrangian leads to

$$\begin{aligned}
\mathscr{L}_a^X = &-\frac{1}{4} c_{a\gamma\gamma} \frac{a}{f_a} F_{\mu\nu} \widetilde{F}^{\mu\nu} - \frac{1}{4} c_{a\gamma Z} \frac{a}{f_a} F_{\mu\nu} \widetilde{Z}^{\mu\nu} - \frac{1}{4} c_{aZZ} \frac{a}{f_a} Z_{\mu\nu} \widetilde{Z}^{\mu\nu} \\
&- \frac{1}{2} c_{aWW} \frac{a}{f_a} W_{\mu\nu}^+ \widetilde{W}^{-\mu\nu} - \frac{1}{4} c_{agg} \frac{a}{f_a} G_{\mu\nu}^a \widetilde{G}^{a\mu\nu},
\end{aligned} \tag{16}$$

where the matching is given by

$$c_{a\gamma\gamma} \equiv c_w^2 c_{\widetilde{B}} + s_w^2 c_{\widetilde{W}}, \qquad c_{a\gamma Z} \equiv 2 c_s s_w \left( c_{\widetilde{W}} - c_{\widetilde{B}} \right), \qquad c_{aZZ} \equiv s_w^2 c_{\widetilde{B}} + c_w^2 c_{\widetilde{W}}, \tag{17}$$

$$c_{aWW} = c_{\widetilde{W}}, \qquad c_{agg} = c_{\widetilde{G}}, \tag{18}$$

with $s_w$ and $c_w$ being the sine and cosine of the Weinberg angle, respectively.

The above discussion deals only with tree-level ALP couplings. The impacts of 1-loop contributions and running on ALP phenomenology have been extensively studied in the literature [126,127,157,179,180,184]. Such contributions correct the tree-level coupling and we can define effective 1-loop couplings as

$$c_X^{\text{eff}} \equiv c_X + \delta c_X^{\text{1-loop}}, \tag{19}$$

where $\delta c_X^{\text{1-loop}}$ is the 1-loop contribution and $X$ is either a fermion or a gauge boson. The precise structure of such corrections is in general rather complicated and momentum-dependent. However, in the high momentum transfer limit, $p^2 \gg v^2$, the momentum dependence drops,

EW symmetry gets restored and the relations of Eq. (17) become valid again (see Refs. [157, 180]). This matches our case study as the energy of the process must be large enough to produce two on-shell HNLs. This allows us to estimate the relevance of the corrections by considering the naive scaling of the 1-loop triangle contributions

$$\delta c_X^{\text{1-loop}} \sim \frac{g_i^2}{16\pi^2} c_Y \,, \tag{20}$$

where $X, Y$ are either gauge-bosons or fermions, $i = 1, 2, 3$ selects the largest gauge coupling appearing in the relative Feynman diagrams and the gauge couplings are defined via $g_3 = \sqrt{4\pi\alpha_s}$, $g_2 = e/\sin(\theta_w)$ and $g_1 = e/\cos(\theta_w)$, with $e = \sqrt{4\pi\alpha_{\text{em}}}$. The above estimation matches explicit computations [157,180] up to hypercharge or colour factors which introduce $\mathcal{O}(1)$ corrections. As it will be explained later, such contributions do not change the conclusions of the analysis and therefore we do not report them. A summary of such estimates starting from a single tree-level coupling at a time can be found in Tab. 1.

The description so far refers to ALPs with masses above $\sim 2$ GeV. Indeed, if a sub-GeV ALP has anomalous couplings with gluons, then a more appropriate description is through the Chiral Perturbation theory, which implements mesons and hadrons instead of free quarks (see Refs. [176,185–188]). According to this treatment, the ALP also acquires a tree-level coupling to photons induced by the ALP mixing with the neutral mesons. We will consider this aspect in more detail in the next sections.

As already mentioned above, the goal of this paper is not to review the ALP phenomenology at colliders assuming the most generic ALP EFT Lagrangian. Instead, the focus is studying the consequences of ALP-HNL couplings, extending the work in Ref. [160], where only ALP couplings to gluons and a single HNL have been analysed. We extend that work by including couplings to other fermions and gauge bosons and by considering the presence of two HNLs, which is a requirement to provide a realistic description of the active neutrino masses. Switching on all these couplings at a time would lead, however, to unnecessarily complicated scenarios. On the contrary, we will restrict our analysis to two non-vanishing couplings at the time but include the loop-induced ones to the other fermions and gauge bosons. The advantage is to limit the analysis to a two-dimensional parameter space with the additional dependence on the HNL and ALP masses. On the other hand, as we will discuss later on, this hypothesis is realistic as the ALP production mechanisms are generically dominated by only one coupling. Tab. 1 identifies the different benchmarks we will study in the next sections. In all the cases, we assume a tree-level coupling of the ALP to HNLs, being the main focus of this paper. Moreover, for each scenario, we take only one additional ALP coupling at tree-level and show the 1-loop level generated ones: couplings to gluons in the first line, to $SU(2)_L$ gauge bosons in the second, to hypercharge gauge boson in the third, to quarks in the fourth and leptons in the fifth and last line. Each line thus identifies a specific ALP construction that corresponds to possible UV completions: gluon-philic, $SU(2)_L$-philic, hypercharge-philic, quark-philic and lepto-philic, respectively. The different columns refer to the effective couplings at low energy defined in Eq. (19). For simplicity, the tree-level couplings are just 1 or 1/2 factors, while the 1-loop effective ones are naive estimations that ignore the $\mathcal{O}(1)$ prefactors. Moreover, to avoid cancellations in the induced effective gauge couplings, "−" signs have been added in front of some tree-level fermion coefficients – see Eqs. (14) and (17).

One may wonder if it is reasonable to assume simultaneous Wilson coefficients of $\mathcal{O}(1)$ for the HNL and SM fermions or gauge bosons. A class of UV models with such characteristics can be obtained by modifying the traditional invisible QCD axion models, such as DFSZ [78,79] and KSVZ [80,81]. For example, if the complex scalar field, that originates the axion/ALP after the PQ SSB, couples to the Majorana mass term of the RH neutrinos, then the axion/ALP itself obtains $\mathcal{O}(1)$ couplings to the HNLs, beside the SM fermions. On the other hand, Wilson coefficients of the axion/ALP to gauge bosons are governed by the PQ anomaly and are

Table 1: Five representative benchmarks, each obtained by turning on one tree-level ALP coupling at a time (in addition to $c_N$), and then estimating at the order-of-magnitude level its 1-loop contribution to the other effective couplings, ignoring the $\mathcal{O}(1)$ prefactors from the loop factors. In the last two lines, "−" signs are artificially added in front of some fermions couplings to avoid cancellations in the induced effective gauge couplings.

| Benchmark | Tree-level coup. | $c_N$ | $c_{\widetilde{G}}^{\text{eff}}$ | $c_{\widetilde{W}}^{\text{eff}}$ | $c_{\widetilde{B}}^{\text{eff}}$ | $c_Q^{\text{eff}}$ | $c_u^{\text{eff}}$ | $c_d^{\text{eff}}$ | $c_L^{\text{eff}}$ | $c_\ell^{\text{eff}}$ |
|---|---|---|---|---|---|---|---|---|---|---|
| BM($\widetilde{G}$) | $c_{\widetilde{G}}$ | 1 | 1 | 0 | 0 | $-\frac{g_3^2}{16\pi^2}$ | $\frac{g_3^2}{16\pi^2}$ | $\frac{g_3^2}{16\pi^2}$ | 0 | 0 |
| BM($\widetilde{W}$) | $c_{\widetilde{W}}$ | 1 | 0 | 1 | $\frac{g_2^2}{16\pi^2}$ | $\frac{g_2^2}{16\pi^2}$ | 0 | 0 | $\frac{g_2^2}{16\pi^2}$ | 0 |
| BM($\widetilde{B}$) | $c_{\widetilde{B}}$ | 1 | 0 | $\frac{g_1^2}{16\pi^2}$ | 1 | $-\frac{g_1^2}{16\pi^2}$ | $\frac{g_1^2}{16\pi^2}$ | $\frac{g_1^2}{16\pi^2}$ | $-\frac{g_1^2}{16\pi^2}$ | $\frac{g_1^2}{16\pi^2}$ |
| BM($q$) | $c_{u,d}-c_Q$ | 1 | $\frac{g_3^2}{16\pi^2}$ | $\frac{g_2^2}{16\pi^2}$ | $\frac{g_1^2}{16\pi^2}$ | $-\frac{1}{2}$ | $\frac{1}{2}$ | $\frac{1}{2}$ | 0 | 0 |
| BM($\ell$) | $c_\ell-c_L$ | 1 | 0 | $\frac{g_2^2}{16\pi^2}$ | $\frac{g_1^2}{16\pi^2}$ | 0 | 0 | 0 | $-\frac{1}{2}$ | $\frac{1}{2}$ |

therefore loop-suppressed. A simple toy model with these properties can be found e.g. in Ref. [99]. Proven that reasonable UV models exist, we choose to be agnostic and adopt an EFT description.

The following two sections contain our phenomenological analysis. Sect. 3 extend the study in Ref. [160] considering only one HNL, but with a broader range of production mechanisms and decays channels, depending on the benchmarks identified in Tab. 1. Two definite experimental setups are considered: the High-Luminosity LHC and a proposed muon collider design. The High-Luminosity LHC is set to be a major upgrade of the LHC, due to start running in the late 2020s, that will increase its integrated luminosity ten-fold to around $3\,\text{ab}^{-1}$ after a decade of operation while maintaining the same centre-of-mass energy of 13.6 TeV (or possibly slightly higher). Contrary to the High-Luminosity LHC, a muon collider would represent a significant departure from tried-and-tested technologies, and as such its feasibility is not guaranteed. Nonetheless, we chose to include the (somewhat optimistic) design proposed in Ref. [163] due to its interesting (and complementary) physical reach when the ALP couples to electroweak bosons. The nominal parameters for this design are a 10 TeV centre-of-mass energy and an integrated luminosity of $10\,\text{ab}^{-1}$.

Subsequently, Sect. 4 will focus on the two HNL setup. In all generality, we can define $c_N$ as

$$c_N \equiv \begin{pmatrix} c_{N,11} & c_{N,12} \\ c_{N,12} & c_{N,22} \end{pmatrix}, \tag{21}$$

and different assumptions can be made:

$c_{N,11} \gtrsim c_{N,22} \gg c_{N,12}$. In this case, the relevant phenomenology is the one described in Sect. 3, with $N_1$ being the lightest HNL.

$c_{N,12} \gg c_{N,11}, c_{N,22}$ or democratic texture. In this setup, the ALP may decay into $N_1 + N_2$ and the heavy HNL may subsequently decay into $N_1 + a$, giving rise to a cascade process. This topology gives a significantly different signal with respect to the JALZ and it is the main focus of Sect. 4.

$c_{N,22}$ dominance. This case is pretty involved as the corresponding phenomenology depends on the relative strength of $c_{N,22}$ with respect to $c_{N,11}$ and the mass splitting between the two HNLs. In the simplified scenario in which $c_{N,11}=0$, then the ALP may decay into two $N_2$ that subsequently decay into two $N_1$, either emitting an ALP or a $Z$ gauge boson. We will comment on this scenario at the of Sect. 4.

Table 2: Total cross section for the processes $pp \rightarrow NN + X$ (where $X = \varnothing, \text{jets}, \ell^+\ell^-, \gamma, \nu\nu$ denotes the main particles co-produced along with the two HNLs) at the LHC and $\mu^+\mu^- \rightarrow NN + X$ at a 10 TeV muon collider, for the five representative benchmarks introduced in Tab. 1 assuming $c_N = 1$, with an HNL mass of $M_N = 400$ or 1600 GeV and $f_a = 10$ TeV.

| Benchmark | $\sigma_{400}^{\text{LHC}}$ [pb] | $\sigma_{1600}^{\text{LHC}}$ [pb] | $\sigma_{400}^{\text{MuC}}$ [pb] | $\sigma_{1600}^{\text{MuC}}$ [pb] |
|---|---|---|---|---|
| BM($\widetilde{G}$) | $2.0 \times 10^{-4}$ | $2.4 \times 10^{-6}$ | $\approx 0$ | $\approx 0$ |
| BM($\widetilde{W}$) | $1.9 \times 10^{-7}$ | $1.4 \times 10^{-8}$ | $2.4 \times 10^{-6}$ | $4.9 \times 10^{-6}$ |
| BM($\widetilde{B}$) | $5.0 \times 10^{-8}$ | $3.7 \times 10^{-9}$ | $2.1 \times 10^{-6}$ | $5.6 \times 10^{-6}$ |
| BM($q$) | $1 \times 10^{-8}$ | $9 \times 10^{-11}$ | $2 \times 10^{-11}$ | $4 \times 10^{-11}$ |
| BM($\ell$) | $1 \times 10^{-12}$ | $1 \times 10^{-13}$ | $2 \times 10^{-11}$ | $4 \times 10^{-11}$ |

## 3 Single-HNL case

This section is devoted to extending the study in Ref. [160]. The setup is the SM plus an ALP and a single HNL and the relevant effective Lagrangian is the one in Eq. (13), with the following conditions: i) all the charged fermions are mass eigenstates, consistent with the universality assumption; ii) $c_N$ is a number and does not have any flavour structure, as only one HNL is considered in this section; iii) the Wilson coefficients are the effective ones including the 1-loop contributions, defined in Eq. (19). It is useful to underline the ALP couplings with the neutral leptons:

$$\mathscr{L}_a \supset \frac{\partial_\mu a}{f_a} \left\{ c_L \overline{\nu_L} \gamma^\mu \nu_L + c_N \overline{N_R} \gamma^\mu N_R + (c_L + c_N) \left[ \overline{\nu_L} \gamma^\mu \Theta N_R^c + \overline{N_R} \gamma^\mu \Theta^\dagger \nu_L^c \right] \right\}. \tag{22}$$

The last two terms (in square brackets) are suppressed with respect to the previous two, since they contain the $\Theta$ factor, thus we will ignore them.

In what follows, we first discuss the different ALP production mechanisms, then the various HNL decay channels to conclude with the results of the numerical analysis.

### 3.1 Production processes

In traditional HNL models, producing the HNLs can be extremely challenging, due to the tiny active-sterile mixing angles $\Theta$. However, as discussed in Ref. [160], the new ALP-HNL coupling enables new production processes mediated by an off-shell ALP ($a^* \rightarrow NN$), which quickly become dominant for sufficiently small active-sterile mixing angles. Meanwhile, the ALP can itself be produced through a variety of processes. $s$-channel production, mediated by the shift-preserving interaction with fermions $\partial_\mu a \bar{f} \gamma^\mu \gamma_5 f$, is present both at proton and muon colliders, but is highly suppressed due to the derivative coupling and the resulting proportionality to the fermion mass. Production through the vector boson fusion (VBF) and ALPstrahlung processes (and variations thereof), both mediated by the anomalous operators $aX\widetilde{X}$, does not suffer from this suppression.

Since each Lagrangian term generates additional effective couplings $c_X^{\text{eff}}$ at the 1-loop level, it is not clear a priori which production process is dominant for a given tree-level coupling $c_X$. To more accurately estimate the relative contributions of the various interactions, we list in Tab. 2 the total cross sections estimated using MadGraph — at the order-of-magnitude level — at the LHC and a 10-TeV muon collider, as a function of the tree-level couplings $c_X$. The ALP scale $f_a$ is fixed at 10 TeV as a title of example. For each $c_X$, we estimate the resulting effective

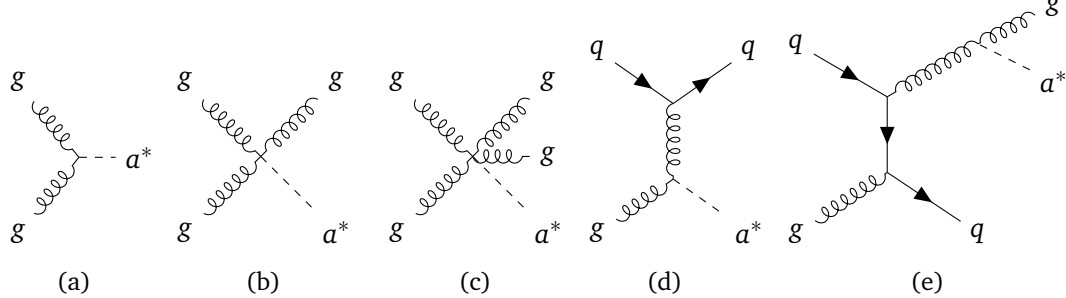

Figure 1: Some representative Feynman diagrams for the dominant ALP production mechanisms mediated by the QCD interaction at the LHC. Off-shell ALPs are produced through VBF, $t$-channel gluon exchange, ALPstrahlung, and variants thereof. Only diagrams with independent initial and final states are shown. Additional $t$-channel and ALPstrahlung topologies arise when considering different internal propagators.

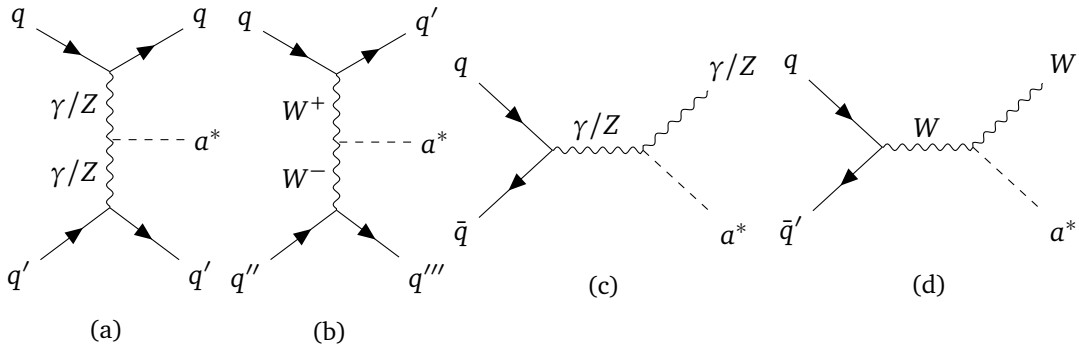

Figure 2: Feynman diagrams involved in the ALP production via the electroweak interaction at the LHC. Off-shell ALPs are produced through VBF (dominant, diagrams **a** and **b**) and ALPstrahlung (very subdominant, diagrams **c** and **d**).

couplings at 1-loop, using only the $g^2/16\pi^2$ loop factor and ignoring the process-dependent prefactor. This rough approximation proved to be sufficient to identify the dominant processes. The full numerical details can be found in Appendix A.

$s$-channel production through fermionic operators was found to be negligible in all considered cases. At the LHC, gluon fusion (mediated by $c_{\widetilde{G}}^{\text{eff}}$) dominates whenever tree-level couplings to quarks or gluons are present, closely followed by a multitude of gluon fusion and ALPstrahlung variants (with the initial gluon(s) emitted by other partons; see Fig. 1). These processes result in intermediate states that consist of two HNLs and up to two jets. If the ALP couples only to leptons or electroweak bosons, the dominant process is instead electroweak VBF (mediated by $c_{\widetilde{W}}^{\text{eff}}$ or $c_{\widetilde{B}}^{\text{eff}}$), that produces an intermediate state with two HNLs and two jets (see Figs. 2a and 2b), while the ALPstrahlung process (Figs. 2c and 2d) is very subdominant. Finally, at the muon collider, the electroweak ALPstrahlung process (Fig. 3c) dominates in all but the BM($\widetilde{G}$) case, with a subleading but sizeable contribution from electroweak VBF (Figs. 3a and 3b). It leads to a variety of final states with zero total charge and consisting of two HNLs plus two neutrinos, one photon, two jets ($q\bar{q}$) or $\ell^+\ell^-$ (with $q\bar{q}$ and $\ell^+\ell^-$ coming mostly from an on-shell $Z$). In the BM($\widetilde{G}$) case, the leading 1-loop contribution is mediated by the ALP-quark interaction and therefore highly suppressed. A precise analysis of this case would require taking into account higher-loop contributions from $c_{\widetilde{G}}$ to $c_{\widetilde{W}}^{\text{eff}}$ and $c_{\widetilde{B}}^{\text{eff}}$; here, we will simply consider this cross section as negligible.

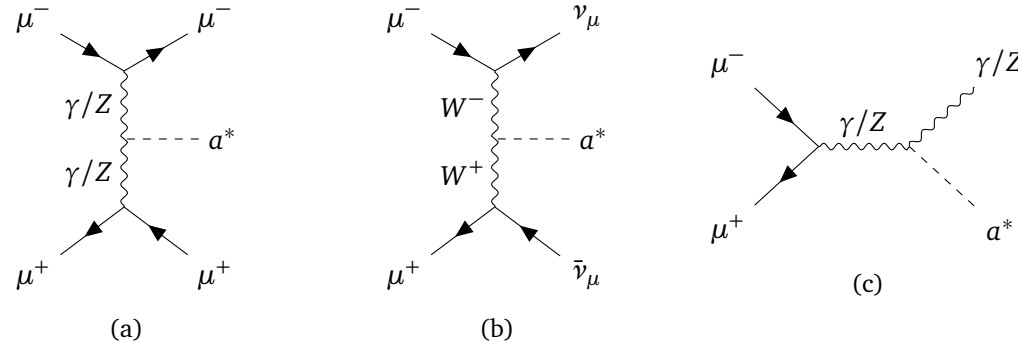

Figure 3: Dominant Feynman diagrams for the ALP production via the electroweak interaction at a muon collider. Off-shell ALPs are produced through VBF (diagrams **a** and **b**) and ALPstrahlung (diagram **c**).

The limited number of relevant intermediate states allows us to cluster the models into two main categories: models with "gluon-philic" ALPs, where the dominant production modes are mediated by $aG\widetilde{G}$, and "electroweak-philic" ALPs where they are mediated by $aW\widetilde{W}$ or $aB\widetilde{B}$. The production cross sections of the various intermediate states are summarised for these two cases in Fig. 4, choosing $c_{\widetilde{B}}^{\text{eff}} = c_{\widetilde{W}}^{\text{eff}}$ for the electroweak-philic benchmark. In the gluon-philic case, the $NN$+jets final state is overwhelmingly dominant at the LHC, while the overall signal is negligible at the muon collider. In the electroweak-philic case, the $NN$+dijet final state largely dominates at the LHC, while the signal at the muon collider involves a wider variety of final states, with the most important being $NN$+invisible, $NN + Z$ (with $Z \to$ dijet) and $NN$+monophoton. Note that the ALP mass does not affect the production yields. However, we will focus on the case $m_a > 0.1$ GeV, since lower masses are ruled out by astrophysical and cosmological constraints for the values of $f_a$ we consider in the analysis [149].

## 3.2 HNL decays

In the case considered in this section, where a single HNL couples to the ALP, the only allowed HNL decays are those mediated by its mixing with the active neutrinos. For HNL masses $M_N$ above the electroweak scale, they consist predominantly of decays into an on-shell electroweak boson and a lepton, namely $N \to W\ell$, $Z\nu$, $h\nu$, in the approximate ratios $2 : 1 : 1$. An additional decay channel is $N \to a\nu$, whose amplitude is proportional to $M_N\Theta/f_a$, which is a stronger suppression with respect to the previous ones. The decay that is most interesting to us is $N \to W\ell$, with the $W$ subsequently decaying hadronically, since it allows reconstructing the mass of the HNL, thus potentially reducing the background. The relative flavour fraction $e : \mu : \tau$ is then approximately proportional to the mixing pattern $|\Theta_e|^2 : |\Theta_\mu|^2 : |\Theta_\tau|^2$.

Just like $N \to Z\nu, h\nu$, decays into $W\tau$ can be problematic since tau leptons are unstable and all of their decays involve at least one neutrino $\nu_\tau$, thus preventing us from directly reconstructing the HNL mass. Therefore, whether decays that involve taus can be used entirely depends on the analysis strategy and, like in many HNL searches, there is no guarantee that HNLs that mix predominantly with $\nu_\tau$ can be efficiently studied using the processes considered here. This introduces a slight dependence of the sensitivity on the precise mixing pattern (see Refs. [22, 189] for examples highlighting the importance of the relative mixings in conventional HNL searches, and Ref. [160, Sect. IV] for an estimate with the JALZ process that assumes fixed tau reconstruction/identification efficiencies).

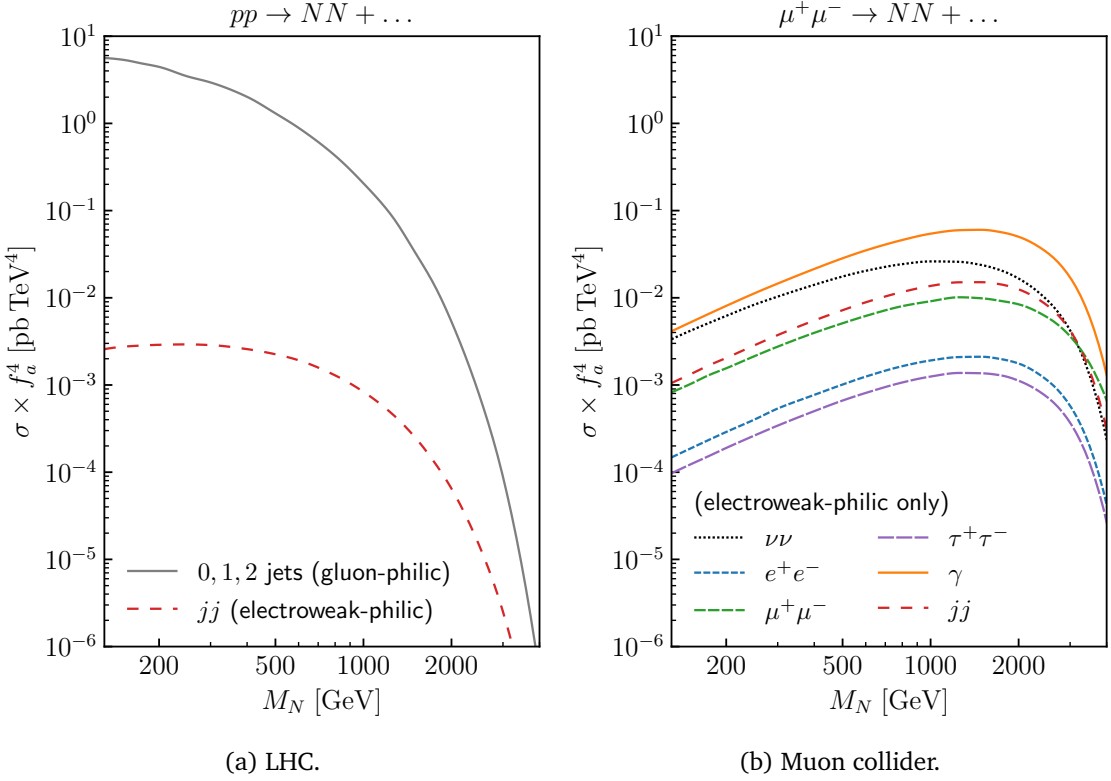

Figure 4: Production cross sections (in units of $f_a^4$) for a pair of HNLs mediated by an off-shell ALP, at the LHC (13.6 TeV) and a future 10 TeV muon collider, for models involving a gluon-philic or electroweak-philic ALP (see the text for details).

The small HNL mixing angles result in widths that are much smaller than the HNL mass, allowing us to employ the narrow-width approximation to factor the overall process into the HNL production and the two HNL decays:

$$\sigma_{\text{overall}} = \sigma_{\text{production}} \times (\mathcal{B}_{\text{decay}})^2, \tag{23}$$

where $\mathcal{B}_{\text{decay}}$ denotes the branching ratio into all the decays of interest (e.g. $N \to \ell jj$). The case of intermediate non-resonant HNL accompanied by ALP emission is found to be significantly suppressed compared to the resonant process involving only on-shell HNLs, at least for HNL widths consistent with the generic Type-I Seesaw.

## 3.3 Experimental signatures

We will consider two experimental setups in this study: the High-Luminosity LHC, with centre-of-mass energy 13.6 TeV and integrated luminosity 3 ab$^{-1}$, and a tentative $10-$ TeV muon collider [163] with an integrated luminosity of 10 ab$^{-1}$. In both experiments, we consider the signature consisting of two HNLs each decaying to a fully reconstructible final state consisting of one lepton $\ell$ and a $W$ boson, itself decaying hadronically to two (possibly-collimated) jets — referred to as the "JALZ" process in Ref. [160] and depicted in Fig. 5. The two leptons can, in general, have any combination of flavours (including $\tau$). When considering a generic Type-I Seesaw, they can additionally have any combination of charges, due to the Majorana nature of HNLs and the resulting lepton number violation. On the contrary, lepton number violating effects will be suppressed in LN-protected Seesaws at large mixing angles, where the quasi-Dirac HNLs decay before the onset of oscillations [49, 190–192], producing opposite-charge leptons.

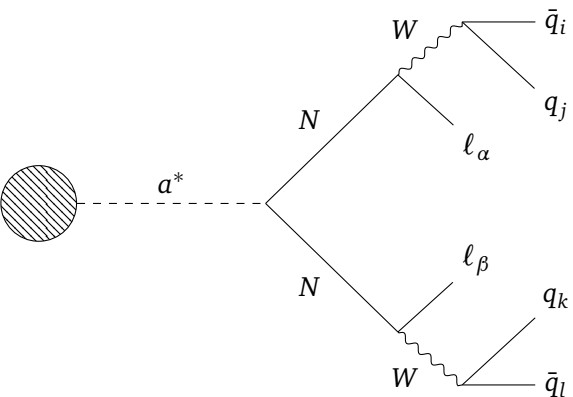

Figure 5: Feynman diagram for the decay part of the JALZ topology. The blob represents any production process from Figs. 1 to 3. The two HNLs $N$ and the two $W$ bosons are on-shell. $\alpha, \beta$ and $i, j, k, l$ are respectively lepton and light quark flavour indices.

While the decays are the same in all experiments for both gluon-philic and electroweak-philic ALPs, the production mechanisms differ significantly, as previously discussed in Sect. 3.1. At the LHC, a gluon-philic ALP can be produced alongside any number of jets, which can complicate the event reconstruction and increase the combinatorial background. However, those can in principle be reconstructed, resulting in no missing transverse energy. Furthermore, the process with zero additional jets is dominant, so one can request exactly 4 jets with little loss in sensitivity. Meanwhile, electroweak-philic ALPs can only be produced with at least two additional jets, also with no missing transverse energy. The signature is more complicated at the muon collider, due to a non-trivial mix of production processes involving fully reconstructible $(\gamma, jj, e^+e^-, \mu^+\mu^-)$, partially reconstructible $(\tau^+\tau^-)$ and invisible $(\nu\nu)$ particles. Since the momenta of the incoming muon and anti-muon are expected to be well-known at the muon collider, the former processes should allow one to fully reconstruct the event kinematics, while in the latter two, neutrinos would result in some missing momentum. While this would prevent the kinematics from being fully reconstructed (which otherwise provides an extra consistency check for the signal), this does not affect the reconstruction of the invariant masses of the two HNLs. Therefore we include these production mechanisms in the total cross section, at least for the non-cascade case discussed in the present section. Note, however, that the exact ratio of $c_{\widetilde{W}}$ and $c_{\widetilde{B}}$ (taken to be $1:1$ in the electroweak-philic benchmark) will affect the relative ordering of the various production processes at the muon collider.

The expected signal yields at the High-Luminosity LHC and the 10-TeV muon collider are shown as a function of the HNL mass in Fig. 6, assuming a signal efficiency of $\epsilon_{\text{sig}} = 1$ and no background. In any realistic search, the signal efficiency will be smaller than 1, and the yields shown in Fig. 6 should be multiplied by $\epsilon_{\text{sig}}$. Equivalently, the ALP scale $f_a$ that can be probed should be replaced by $(\epsilon_{\text{sig}})^{1/4} f_a$. The signal yields (and the resulting sensitivity) are affected by various sources of uncertainty, which are discussed at the end of App. A. On the other hand, a conservative numerical estimation of the background at LHC, considering the gluon-philic case, is presented in App. B.

The main advantage of the JALZ signature is that the on-shellness of the two HNLs and $W$'s severely restricts the invariant masses of various combinations of final-state particles, resulting in a "smoking gun" signal with four simultaneous mass peaks expected — observable in all the considered experiments. Not only we do expect this very specific kinematic structure to strongly reduce the SM background for opposite-sign same-flavour processes, but it should also help suppress the combinatorial background coming from e.g. pileup, which affects all

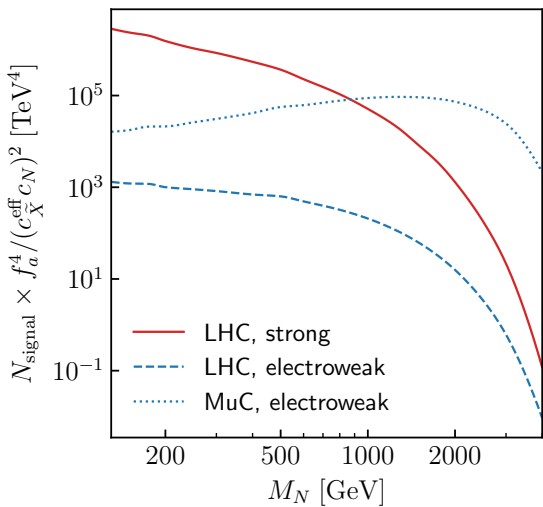

Figure 6: Expected number of signal events excluding opposite-sign same-flavour processes at the high-luminosity LHC ($3\,\text{ab}^{-1}$) and a 10 TeV muon collider ($10\,\text{ab}^{-1}$), normalized by the ALP couplings, and under the no-background hypothesis.

processes. However, for simplicity, we will exclude opposite-sign same-flavour processes from the following analysis, focusing instead on the remaining processes which are less affected by the SM background (a relatively recent analysis can be found in Ref. [193]). In the case of democratic mixing, this incurs a reduction by a factor of $\approx 5/6$ in the number of events. The JALZ signature does not come without challenges, though. Indeed, due to the high mass of the HNLs (especially above $\sim 1$ TeV), each $W$ boson can be significantly boosted, and the two jets resulting from its decay can become collimated. As observed in Ref. [160], such collimated jets could get rejected by standard $\Delta R_{jj}$ cuts, and it might be necessary to treat them as a single large-radius jet, potentially leveraging the jet substructure associated with the $W$ decay. In addition, decays that involve tau leptons are only partially reconstructible, since they necessarily involve at least one neutrino each. This can impede the reconstruction of the $W$ and HNL masses, and reduce the useful signal when the HNL mixes predominantly with $\nu_\tau$. Leptonic tau decays otherwise look like electrons, muons or light jets plus missing momentum, therefore relaxing the missing transverse energy or missing momentum cuts could provide some sensitivity to HNLs mixing with taus, but at the cost of increasing the background. Semileptonic tau decays, on the other hand, would produce additional hadronic activity in an already-busy event, making their detection potentially challenging. In either case, the precise knowledge of the initial state at the muon collider could allow reconstructing the momentum of up to one $\nu_\tau$. Whether collimated jets and missing momentum are truly an issue and affect the final sensitivity will eventually depend on the chosen analysis strategy.

In a conventional "cut-and-count" analysis, the various production mechanisms and kinematic regimes (resolved *vs.* boosted jets) could lead to a multiplication of signal regions and complicated analysis, unless one restricts oneself to the dominant production mechanisms at the expense of sensitivity.

Maybe a simpler way to search for the JALZ process would be to perform a double peak search, analogous to the search proposed in Ref. [194]. Consider all distinct pairings of hadronically-decaying $W_\text{h}$'s with a charged lepton $\ell = e, \mu$. Then, among events with exactly two such pairs, plot/bin for each pairing the invariant masses of the first and second $W_\text{h}\ell$ pair, one on each axis. The JALZ signal would then show up as a peak along the diagonal, and the off-diagonal events/bins can be used to estimate the background (that will necessarily include

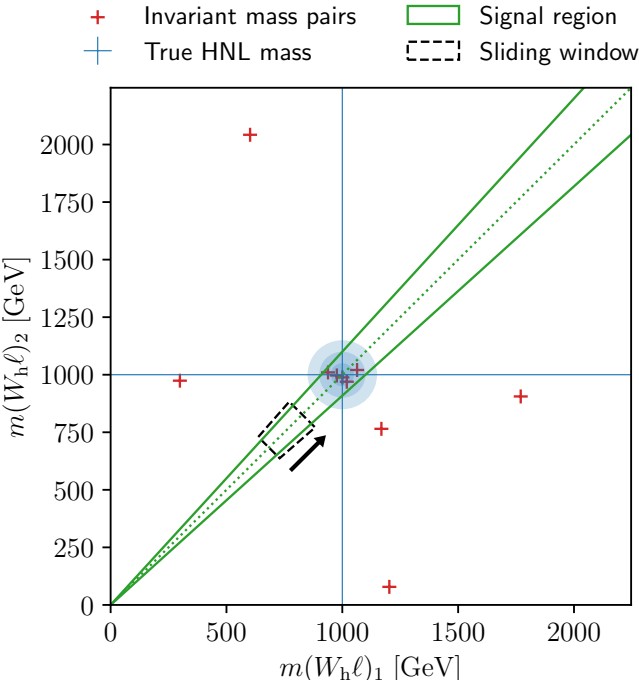

Figure 7: Schematic depiction of the double peak search described in the text. The mass of each $W_{\mathrm{h}}\ell$ pair is plotted on a different axis, and each point (red +) represents a possible pairing of hadronically-decaying $W$'s with leptons in an event with exactly two such pairings. The true HNL mass (taken to be 1 TeV in this example) is shown in blue, along with the distribution resulting from the finite resolution of the experiment. A sliding window approach is employed within the signal region (in green) which encompasses the diagonal (dotted).

a combinatorial component due to the sum over pairings) in a data-driven way (note, however, that events containing a single $W_{\mathrm{h}}\ell$ cannot be used for the background estimation, since they may be contaminated by signal where one of the HNL decays is only partially-reconstructed). Since this method only searches for decaying HNLs, it is agnostic to the production mechanism. However, it will not be sensitive to the $\ell = \tau$ case due to the missing momentum. This method is depicted schematically in Fig. 7.

Finally, the JALZ process can also be searched for using machine-learning-based anomaly-detection methods (ideally tuned to this type of signal). This might allow searching more efficiently for partially-reconstructible HNL decays (such as $N \rightarrow W_{\mathrm{h}}\tau$, $N \rightarrow \ell(W \rightarrow \ell \nu)$ or $N \rightarrow Z/h\nu$), for which strict reconstruction of invariant masses cannot be performed. However, it also requires having a good understanding of the "simulation gap" and associated systematic uncertainties.

Before moving to the description of the numerical results, we comment on the expected SM background. From one side, a final state with opposite-sign different-flavour leptons could be produced by top pair decays, although this potential background could be avoided by requiring a fully reconstructable signal, as in our case. On the other side, there are a series of processes that could contribute to the background of same-sign different-flavour lepton final state. Moreover, one should consider the possibility of particle misidentification and/or an erroneous charge identification and this has already been considered in the literature: one can see Ref. [18] for the HNL case. In the following section, we present the expected number of signals in the background-free case for the same-sign same-flavour lepton final states as

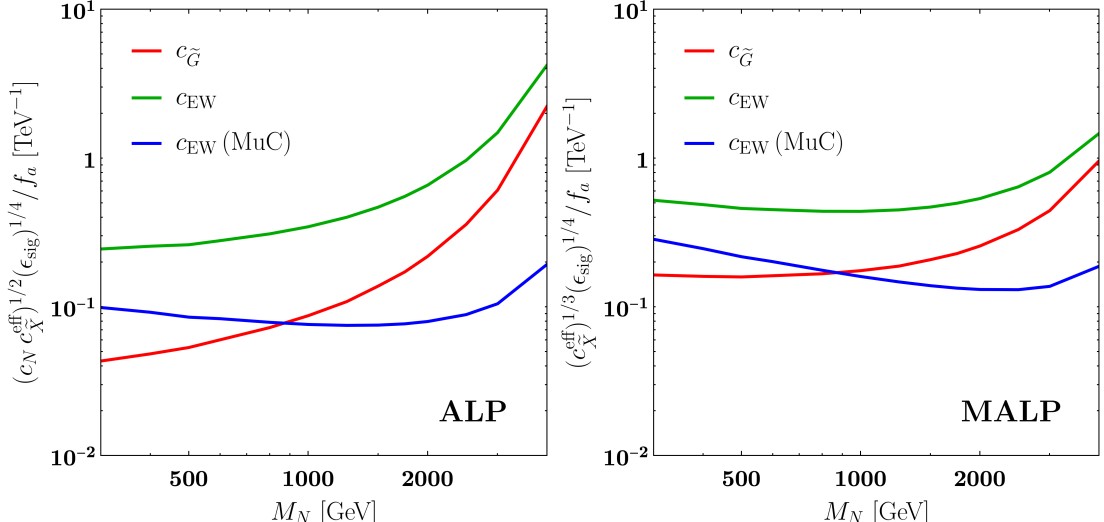

Figure 8: Values of the strong and EW ALP couplings necessary to obtain 3 events with a luminosity of 3 ab$^{-1}$. The Majoron-like case (right plot) is obtained by imposing $M_N = c_N f_a / \sqrt{2}$.

a function of the ALP couplings. We conservatively estimate the background's impact on the JALZ signal in App. B by adapting the results presented in Ref. [18]. We conclude that, for HNL masses up to 1700 GeV, the constraints may weaken by a factor $\sim 6$ (see Fig. (13)). However, notice that such a conclusion is pessimistic as it naively rescales the background for the single HNL case without employing an optimized strategy such as the one presented in Fig. 7, which is likely to greatly reduce the background.

### 3.4 Results

The JALZ process provides strong sensitivity to ALPs coupling to HNLs, as shown in Fig. 8 in the form of lines that corresponds to the observation of 3 events, equivalently to the 95% C.L. sensitivity under the hypothesis of the absence of background.

The left plot shows the bounds for an ALP where no correlation between the HNL mass $M_N$ and its coupling $c_N$ are assumed. The bounds are obtained on the combination $(c_N c_X^{\text{eff}})^{1/2}/f_a$, by using the results in Fig. 6, that assumed all the possible processes, expect for the same-flavour opposite-sign final lepton channels. To be noticed that the results are independent from the HNL mixing pattern as long as $N \to W_h \tau$ decays are included. Assuming Wilson coefficients of $\mathcal{O}(1)$, the bounds generically impose $f_a \gtrsim 1$ TeV and get to $f_a \gtrsim 10$ TeV in the lower $M_N$ range. The bound on $f_a$ gets relaxed as the Wilson coefficients get smaller. However, the JALZ constraints do not depend on the Wilson coefficients linearly but scale with their square root, thus mitigating their impact on $f_a$. This dependence becomes particularly important when the Wilson coefficients are generated at loop-level and are thus expected to be $\delta c_X^{\text{1-loop}} \sim \mathcal{O}(10^{-2})$. Additionally, excluding tau leptons from the final state or imposing more realistic efficiencies on them would also reintroduce a weak dependence on the HNL mixing pattern $\Theta_e : \Theta_\mu : \Theta_\tau$ but, again, its impact on $f_a$ would be limited since the limit scales as the fourth root of the signal efficiency. This justifies *a posteriori* our choice to allow tau leptons in the final state in order to simplify the presentation.

The right plot of Fig. 8 shows the same bounds but enforcing the relation $M_N = c_N f_a / \sqrt{2}$. This removes the $c_N$ dependence completely and recasts the bounds as a function of the ALP-SM couplings only. Such a relation naturally arises in Majoron models where the Majorana mass is generated dynamically via the vev of a complex scalar field; we refer to such a case

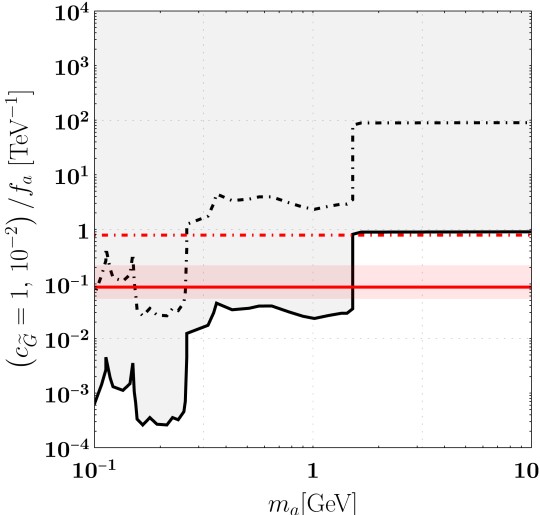

Figure 9: Comparison of the bounds obtained from gluons (red) to the existing flavour constraints from Ref. [127]. The red band spans the values of the HNL mass $M_N \in [0.5, 2]$ TeV $c_N = c_{\widetilde{G}} = 1$. Fixing $M_N = 1$ TeV, the red solid(dot-dashed) line corresponds to $c_{\widetilde{G}} = 1 (c_{\widetilde{G}} = 10^{-2})$.

as *Majoron-ALP (MALP)*. In this case, the bounds can be imposed independently on the value of $c_N$ on the combination $(c_X^{\mathrm{eff}})^{1/3}/f_a$. The MALP bounds are of the same order of magnitude as in the ALP case, but scale with the cubic root of the Wilson coefficients, thus strengthening their relevance in case of loop-suppression.

To appreciate the quality of the bounds, we compare the ALP-case results with the literature in Figs. 9 and 10. Such bounds are typically reported on $c_X/f_a$. Given the different scaling on the Wilson coefficient and the presence of $c_N$, a direct comparison is not possible. For the sake of illustration, we, therefore, make some simplifications: we assume $c_N = c_X = 1$ and use $M_N = 1$ TeV as a benchmark. Different values of the Wilson coefficients can be implemented straightforwardly by performing appropriate rescaling. We consider the ALP flavour constraints summarised in Ref. [127]. For $c_{\widetilde{B},\widetilde{W}}$, we also include the bounds extracted from non-resonant ALP searches in vector-boson scattering [154]. In both cases, the bounds are derived assuming the presence of a single coupling at a time. The presence of multiple couplings, e.g. $c_{\widetilde{G}}$ and $c_{\widetilde{B},\widetilde{W}}$, can lead to stronger bounds, but make comparison impossible; we, therefore, do not include them in the plots.

In Fig. 9, we first report bounds on $c_{\widetilde{G}}$. Their dependence on the exact value of $M_N$ is shown as a red-shaded area, while the solid line represents the benchmark $M_N = 1$ TeV. As can be seen, the new limits are of the same orders of magnitude as the ones in the literature for $0.1$ GeV $\lesssim m_a \lesssim 1$ GeV but become about 10 times stronger for $m_a \gtrsim 1$ GeV. However, if the Wilson coefficients get suppressed, the JALZ bounds scale more favourably. To exemplify the impact of the different scaling on the bounds, we show as dot-dashed lines the case $c_{\widetilde{G}} = 10^{-2}$. As can be seen, the new limits start dominating in the full $m_a$ range. The situation becomes even more dramatic for smaller values of $c_{\widetilde{G}}$.

The comparison with the other couplings, $c_{\widetilde{B},\widetilde{W}}$ and $c_{Q,L}$, can be seen in Fig. 10. The JALZ limits dominate by about a factor of 100 on existing bounds of $c_{\widetilde{B}}$, while they are of the same order of magnitude for $c_{\widetilde{W}}$ and are generically subdominant for the fermionic couplings. Being the JALZ signal non-resonant for what concerns the ALP, the flatness of the limits on $m_a$ always allows it to become stronger than currents bound in specific $m_a$ ranges. This is more evident when looking at the limits on $c_{Q,L}$ where the new bounds dominate for $m_a \gtrsim 5$ GeV. Once

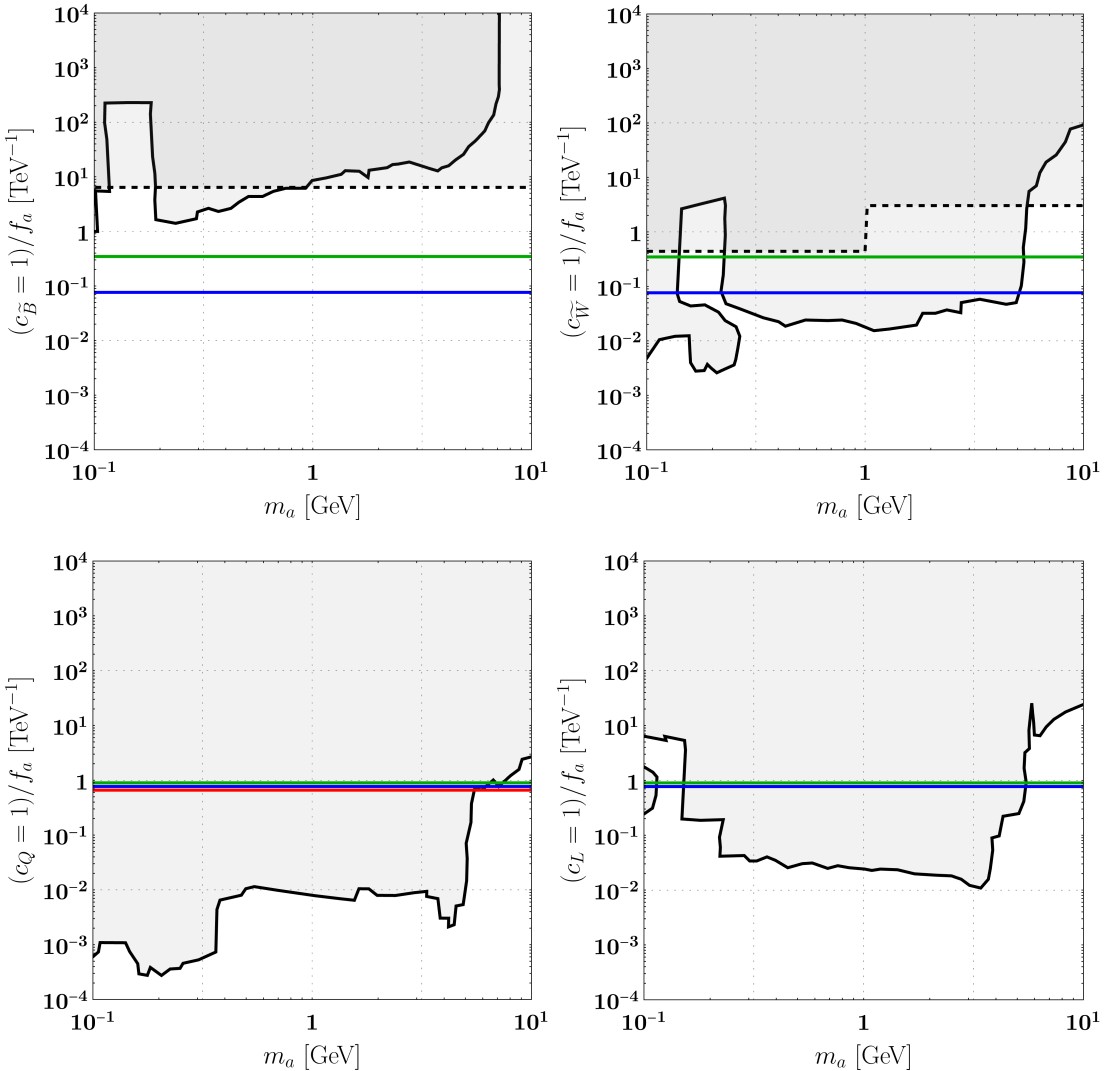

Figure 10: Comparison of the bounds obtained from gluons (red), EW at LHC (green) and EW at muon-collider (blue) to the existing flavour constraints [127] (solid black), non-resonant ALP searches in vector-boson scattering and missing energy studies from $Z/W^{\pm}$ decays at LHC [154,195] (dashed black). The mass of the HNL has been set to $M_N = 1$ TeV and, to allow for a comparison, each coupling to $c_N = c_X = 1$. Notice that the bounds on the quark and lepton couplings, $c_Q$ and $c_L$, are to be taken as order of magnitude estimate as we have not included model dependent $\mathcal{O}(1)$ factors such as hypercharges or colour factors in their estimations.

again, notice that if the Wilson coefficients get suppressed, the JALZ bounds weaken at a much slower rate compared to current bounds, thus allowing them to become competitive or even dominating in such scenarios.

# 4 Two-HNL case

## 4.1 Processes

In Sect. 3, different production mechanisms for ALPs that could potentially be relevant at present and future particle colliders have been studied, as well as their coupling to a single HNL. The present section will focus on the main phenomenological differences associated with the inclusion of two HNLs, which is the minimum number required (although not necessarily sufficient) by the observation of two non-zero mass splittings in neutrino oscillation experiments.

The first phenomenological difference we encounter concerning the single-HNL case is that the coupling $c_N$ is no longer a single parameter. With the inclusion of the second HNL, $c_N$ becomes a $2 \times 2$ symmetric matrix encoding three different couplings: $a-N_1 N_1$, $a-N_2 N_2$ and $a-N_1 N_2$, where $N_2$ and $N_1$ are the heavier and lighter HNLs, respectively. In what follows, we will assume a democratic texture of $c_N$, that is with all the entries in the same ballpark. We will relax this assumption and investigate the consequences at the end of this section.

The other phenomenological impact of the addition of the second HNL is the introduction of cascade decays, in which the heavier HNL $N_2$ decays into the lighter one $N_1$, producing an on-shell ALP in the process. This cascade decay is only kinematically allowed if the mass splitting of the HNL pair is at least the mass of the ALP $m_a$. If the mass splitting is smaller, $N_2$ will instead undergo the same decays as $N_1$, mediated by the mixing between HNLs and neutrinos.

Various models, such as the LSS and the ISS, predict a mass splitting between $N_2$ and $N_1$ at the order of the mass splittings observed in neutrino oscillations: $\Delta M \lesssim$ eV. The ALP considered in this work is significantly heavier than this value, with a lower limit of 0.1 GeV, and consequently, no cascade process can take place in these Seesaw contexts. Instead, $N_2$ and $N_1$ can be studied similarly to the JALZ topology described in Tab. 2. Notice that, due to the small value of $\Delta M$, the invariant mass peaks of the two HNL species cannot be experimentally resolved and the introduction of $N_2$ affects the total cross section for the JALZ process $pp \to 2\ell 4q$: $\sigma_{2\text{HNL}} \cong 4\sigma_{1\text{HNL}}$, since there are now three decay channels ($a^* \to N_1 N_1, N_2 N_2, N_1 N_2$) and the cross-section associated to the production of $a^* \to N_2 N_1$ is expected to be twice the one for $a^* \to N_i N_i$, due to the absence of a symmetry factor. Furthermore, the approximate LN conservation, which characterises these constructions, may suppress the cross sections of LN violating processes [9–12,169–171,196–203], while enhancing the cross sections of LN conserving ones. On the other hand, one may worry that coherent HNL oscillations [49,190–192,204–214], that do represent a source of LN breaking, could further complicate this picture, but, due to unitarity, they do not affect the total cross-section. In summary, in LN-protected Seesaw mechanisms, such as the LSS and ISS, the two HNLs are experimentally indistinguishable from each other and the three ALP-HNL couplings, $c_{N,11}, c_{N,12}, c_{N,22}$, cannot be studied individually.

On the other hand, generic Type-I Seesaw models and the ESS realisation allow an arbitrary mass splitting between the HNL pair, allowing cascade processes to occur. The $N_2 \to N_1 a$ process is now kinematically viable and turns out to be the dominant decay channel: indeed, the traditional HNL decays into SM particles are also viable, but they are produced via the mixing with the active neutrinos, therefore proportional to $\Theta$ that is parametrically suppressed compared to $c_N$. This opens up the possibility to study the properties of $N_2$ and, in particular, of the coupling $c_{N,12}$ that enters into the cascade processes at tree-level. Regarding mass reconstruction, the mass of $N_1$ is generally reconstructible (unless it decays through the $\nu_\tau$ mixing), as previously discussed for the single-HNL case in Sect. 3.2 and 3.3. However, at the LHC, the mass of the heavier HNL $N_2$ will only be reconstructible if the additional ALP emitted in the cascade decays within the detector into detectable particles. In the case of the

Table 3: Total cross section expected at the LHC for the process $pp \rightarrow N_2 N_1 X \rightarrow (N_1 a) N_1 X$ and at a 10 TeV muon collider for the process $\mu^+ \mu^- \rightarrow N_2 N_1 X \rightarrow (N_1 a) N_1 X$, with $X = \varnothing$, jets, $\ell^+ \ell^-$, $\gamma$, $\nu\nu$, with each $N_1$ subsequently decaying as $N_1 \rightarrow \ell j j$. The HNLs masses are fixed to $M_{N_1} = 200$ GeV and $M_{N_2} = 400$ GeV, and $f_a = 10$ TeV.

| Benchmark | $\sigma^{\text{LHC}}$ [pb] | $\sigma^{\text{MuC}}$ [pb] |
|---|---|---|
| BM($\widetilde{G}$) | $7.5 \times 10^{-5}$ | $\approx 0$ |
| BM($\widetilde{W}$) | $5.7 \times 10^{-8}$ | $4.5 \times 10^{-7}$ |
| BM($\widetilde{B}$) | $1.6 \times 10^{-8}$ | $3.8 \times 10^{-7}$ |
| BM($q$) | $4.1 \times 10^{-9}$ | $3.6 \times 10^{-12}$ |
| BM($\ell$) | $4.5 \times 10^{-13}$ | $3.8 \times 10^{-12}$ |

muon collider, since the initial state information is known, the mass of $N_2$ can be reconstructed even if the ALP is long-lived and escapes the detector, provided that the particles that are co-produced along with the two HNLs are reconstructible and sufficiently distinguishable from the ALP's own decay products.

## 4.2 Experimental signatures

As previously discussed, the phenomenology associated with cascade processes significantly differs from the pure JALZ case. One such difference is the production cross-section. Proceeding similarly to the pure JALZ case, in Tab. 3 we study the expected cross section at the two main colliders under study for the five benchmarks listed in Tab. 1. In this case, the computed process for LHC is $pp \rightarrow N_2 N_1 X \rightarrow (N_1 a) N_1 X$ whereas for the muon collider it is $\mu^+ \mu^- \rightarrow N_2 N_1 X \rightarrow (N_1 a) N_1 X$, with $N_1$'s decaying as $N_1 \rightarrow \ell q \bar{q}'$. The observed variations with respect to Tab. 2 come from i) the impact of parton distribution functions (PDFs) when producing HNLs at different masses (200 GeV and 400 GeV in this case), ii) including the final products of the $N_1$ decay, with $\mathcal{B}(N_1 \rightarrow \ell q \bar{q}') \approx 38\%$, as well as iii) an additional factor of 2 that arises when considering a coupling of the type $a - N_1 N_2$, due to the presence of distinguishable final state fermions.

Another key difference is the dependence on the ALP mass $m_a$. The JALZ topology studied in Sect. 3 involved a single off-shell ALP, making the whole process nearly independent from $m_a$. However, this independence does not hold for topologies including cascade processes, where $m_a$ becomes relevant in two distinct ways: first, as previously discussed, $\Delta M > m_a$ is necessary to enable cascade processes, and second, $m_a$ significantly influences the ALP lifetime, determining whether it decays inside or outside the detector. Long-lived ALPs, typically associated with smaller masses, will decay outside the detector, resulting in missing transverse energy (at the LHC) or missing momentum (at the muon collider) as their experimental signature.[1] Conversely, short-lived ALPs will decay within the detector, producing potentially detectable particles.[2] While $m_a$ is a primary parameter influencing the ALP decay rate and products, its couplings also play a crucial role.

---

[1]Note that, at the muon collider, the production mechanisms involving neutrinos or tau leptons will also feature missing momentum. They will therefore constitute a background to ALPs escaping the detector. In order to discriminate between the cascade and non-cascade processes, one might therefore want to focus on the reconstructible production mechanisms only (by requiring two jets, one $\gamma$ or $\ell^+ \ell^-$).

[2]Note that some of the particles co-produced with the two HNLs at the muon collider may be similar to the ALP decay products; however, their very different kinematics should allow distinguishing them.

Table 4: ALP decay rates and main modes for the most relevant mass regimes: $0.1$ GeV $< m_a < m_{3\pi}$, $m_{3\pi} \leq m_a < 2$ GeV, $m_a \gtrsim 2$ GeV, at a scale of $f_a = 10$ TeV. Three given mass values have been selected for definiteness. For each of them, the different benchmarks shown in Tab. 1 have been considered. For $m_a < 2$ GeV, the benchmark BM($q$) is not considered as the chiral description must be adopted. The dots in the BM($\widetilde{G}$) benchmark refer to decays into three light hadrons, different from three pions.

| ALP mass | Benchmark | $\Gamma_a$[GeV] | Main decay mode |
|---|---|---|---|
| $m_a = 0.1$ GeV | BM($\widetilde{G}$) | $1 \times 10^{-14}$ | $a \to \gamma\gamma$ |
| | BM($\widetilde{W}$) | $4 \times 10^{-14}$ | $a \to \gamma\gamma$ |
| | BM($\widetilde{B}$) | $5 \times 10^{-13}$ | $a \to \gamma\gamma$ |
| | BM($\ell$) | $1 \times 10^{-17}$ | $a \to e^+e^-, \gamma\gamma$ |
| $m_a = 1$ GeV | BM($\widetilde{G}$) | $1 \times 10^{-9}$ | $a \to \pi\pi\pi, \ldots$ |
| | BM($\widetilde{W}$) | $4 \times 10^{-11}$ | $a \to \gamma\gamma$ |
| | BM($\widetilde{B}$) | $5 \times 10^{-10}$ | $a \to \gamma\gamma$ |
| | BM($\ell$) | $4 \times 10^{-12}$ | $a \to \mu^+\mu^-$ |
| $m_a = 2$ GeV | BM($\widetilde{G}$) | $5 \times 10^{-8}$ | $a \to jj$ |
| | BM($\widetilde{W}$) | $4 \times 10^{-10}$ | $a \to \gamma\gamma$ |
| | BM($\widetilde{B}$) | $4 \times 10^{-9}$ | $a \to \gamma\gamma$ |
| | BM($q$) | $3 \times 10^{-11}$ | $a \to jj$ |
| | BM($\ell$) | $9 \times 10^{-12}$ | $a \to \mu^+\mu^-$ |

Regarding the possible decay products, three main mass regimes for $m_a$ can be identified and an example for each of them is considered in Tab. 4. These regimes are mainly defined by the differences in experimental signatures associated with an ALP that predominantly couples to gluons via $c_{\widetilde{G}}$, as this coupling has been shown to offer greater sensitivity at colliders. The first regime corresponds to the case in which $0.1$ GeV $< m_a < m_{3\pi} \simeq 0.42$ GeV, for which the chiral description is required. The decay into two pions is forbidden due to parity violation and the ALP mainly decays into photons, with a coupling that is loop-suppressed. Within this regime, we can also identify the region in which $0.1$ GeV $< m_a < 2m_\mu \simeq 0.21$ GeV. When lepton couplings are considered at tree level, ALP decays into photons or $e^+e^-$, with a Branching Ratio of roughly 50% each. However, above the two muon mass threshold, leptophilic ALP decays are dominated by $a \to \mu^+\mu^-$ since the muon mass is $\sim 200$ times the mass of the electron. The second regime corresponds to $m_{3\pi} \leq m_a < 2$ GeV, in which the main decay mode of gluon-philic ALPs is to three pions or, in general, three light hadrons. The third and last regime corresponds to values of $m_a > 2$ GeV. We can assume quark-hadron duality and the observed experimental signal of a gluon-philic ALP decaying into two gluons is two (possibly collimated) jets. This information is summarised in Tab. 4, which also shows that the ALP lifetime scales with $m_a^{-3}$ as long as the decay process remains unchanged. Although ALP decay products and lifetime affect the observed experimental signal, they have no impact on the total cross section expected for cascade processes.

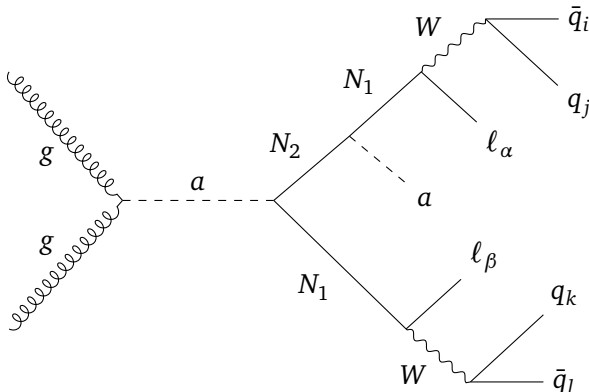

Figure 11: Dominant Feynman diagram for the JALZ topology including cascade process affecting the upper HNL branch. Particles $N_2, N_1$ and $W$'s are produced on-shell.

## 4.3  Results

To study the phenomenology associated with cascade processes in greater detail, we focus on the gluon-philic benchmark (BM($\widetilde{G}$) in Tab. 1) and the $pp \to N_2 N_1 \to (N_1 a) N_1$ signal at the LHC, with gluon fusion being the dominant production mechanism. The corresponding Feynman diagram is depicted in Fig. 11. This combination is characterised by having the largest production cross section among the considered benchmarks, as well as a relatively clean experimental signature (without additional particles being co-produced with the HNLs). At the muon collider, the relevant process is very similar, but with the off-shell ALP being produced by EW gauge boson fusion (see Tab. 3). The analysis and the expected results are therefore qualitatively very similar, allowing us to focus on the LHC case, whose experimental setup is better identified.

Our primary objectives are to estimate the expected sensitivity to this process, identify the corresponding experimental signatures and study how the model parameters impact the total cross-section and possible decay products. To address these questions, we explore, in a scale-independent way, the parameter space associated with the two tree-level couplings, $c_N/f_a$ and $c_{\widetilde{G}}/f_a$, characterising BM($\widetilde{G}$). By considering the case of 3 ab$^{-1}$ integrated luminosity, we indicate the parameter space corresponding to the prediction of 3 and 30 events, in Fig. 12, with different choices of ALP and HNL masses: in Fig. 12a we take $(m_a, M_{N_1}, M_{N_2}) = (0.1, 200, 400)$ GeV; in Fig. 12b we increase the ALP mass up to $m_a = 1$ GeV; in Fig. 12c we increase the HNL masses but keeping the same mass splitting, $(m_a, M_{N_1}, M_{N_2}) = (0.1, 800, 1000)$ GeV; finally, in Fig. 12d we further lift the heaviest HNL mass up to $M_{N_2} = 1600$ GeV with respect to the previous case in order to analyse the dependence on the mass splitting.

The sensitivity to the overall process (regardless of whether the ALP decays within or outside the detector) is estimated for an integrated luminosity of 3 ab$^{-1}$: the solid line refers to the prediction of 30 events; the dashed one to the 3 predicted events case, which corresponds to the 95% C.L. sensitivity in the absence of background. The signal consists in the process $pp \to N_2 N_1 \to (a N_1) N_1$, with each $N_1$ subsequently decaying as $N_1 \to \ell j j$ ($\ell = e, \mu, \tau$) and no opposite-sign same-flavour charged lepton pairs.

Going beyond the overall estimated sensitivity, we then distinguish the regions in parameter space where the ALP is short-lived (shown in pink) or long-lived (blue). We consider the ALP to be long-lived if its boosted lifetime exceeds 1.3 m, which corresponds to the inner radius of the electromagnetic calorimeter (used to reconstruct $a \to \gamma\gamma$) of the CMS experiment [215],

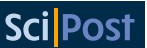

(a) $m_a = 0.1$ GeV, $M_{N_1} = 200$ GeV, $M_{N_2} = 400$ GeV.

(b) $m_a = 1$ GeV, $M_{N_1} = 200$ GeV, $M_{N_2} = 400$ GeV.

(c) $m_a = 0.1$ GeV, $M_{N_1} = 800$ GeV, $M_{N_2} = 1000$ GeV.

(d) $m_a = 0.1$ GeV, $M_{N_1} = 800$ GeV, $M_{N_2} = 1600$ GeV.

Figure 12: Parameter space of ALPs couplings to gluons, $c_{\widetilde{G}}$, and HNLs, $c_{N,12}$, over the scale $f_a$. The latter coupling corresponds to an ALP coupling to $N_1$ and $N_2$. The different figures explore different masses for the ALP, $m_a$, and for the HNLs, $M_{N_1}$ and $M_{N_2}$. The blue (pink) region corresponds to an ALP decaying outside (inside) the detector. The dashed (solid) line refers to the prediction of 3 (30) events for a luminosity of $\mathcal{L} = 3$ ab$^{-1}$.

taking as boost factor the mode of its distribution as reported by `MadAnalysis` [216].[3] The two regions correspond to distinct experimental signals. Because long-lived ALPs predominantly decay outside the detector, events in the blue region will feature missing transverse energy in addition to the decay products of the two light HNLs ($2\ell4j$), whereas in the pink region, the cascade ALP will predominantly decay inside the detector and its decay products ($\gamma\gamma$, hadrons, etc.) may be detectable.

---

[3]This is effectively a monochromatic approximation. Although this rough approximation will be sufficient here, in reality, one should not only consider the full distribution of the boosted lifetime, but also the actual reconstruction algorithm, whose efficiency will likely depend on energy and gradually decrease with the radius.

We finally study the impact of the three masses — parameterised using $m_a, M_{N_1}$ and $\Delta M = M_{N_2} - M_{N_1}$ — on the sensitivity and expected experimental signal associated with cascade processes, by varying each parameter:

$m_a$. Two distinct mass benchmarks have been considered: 0.1 GeV and 1 GeV. At 1 GeV, the ALP is sufficiently short-lived to ensure that it consistently decays within the detector. Higher masses were not considered, as this conclusion would remain unchanged. However, the expected decay products vary according to the mass regime, as detailed in Tab. 4, with hadrons being the predominant decay products of gluon-philic ALPs. For lighter masses, such as 0.1 GeV, the cascade ALP is long lived and decays outside the detector when the ALP-gluon coupling $c_{\widetilde{G}}/f_a$ is sufficiently small, whereas it promptly decays into a pair of detectable photons for larger values of $c_{\widetilde{G}}/f_a$. Given that the ALP masses considered in this study are significantly lower than the ones of HNLs, the ALP lifetime is entirely independent of the $c_N/f_a$ coupling.

$M_{N_1}$. Although the ALP-fermion coupling is proportional to the mass of the fermion, in proton-proton collisions, increasing the HNL masses results in a suppression coming from PDFs, due to the higher parton-parton centre-of-mass energy required to produce both HNLs on shell. This has been deeply discussed in Ref. [160].

$\Delta M$. Larger values of the mass splitting (while keeping $M_{N_1}$ constant) imply a heavier HNL $N_2$. As already discussed, this results in a reduced sensitivity due to PDF suppression. A more subtle effect of this parameter is that larger values of $\Delta M$ generate more energetic ALPs, which can more easily evade the detector due to having a longer lifetime in the lab frame. Consequently, the parameter region where the ALP shows up as missing transverse energy expands.

## 4.4 Comparison between cascade and non-cascade processes

The cascade signal discussed in the present section, and the non-cascade signal discussed in Sect. 3, are complementary. Due to phase-space and PDFs effects, the production cross section is generically larger for lighter HNLs (e.g. $a^* \to N_1 N_1$) than for heavier ones ($a^* \to N_1 N_2$ or $a^* \to N_2 N_2$). This is subject to caveats, as it depends on the size of the corresponding Wilson coefficients ($c_{N,11}$ vs. $c_{N,22}$, $c_{N,12}$) and, ultimately, on the UV model which generates them. Overall, there are three scenarios:

**Dominant diagonal interactions with $c_{N,11} \gtrsim c_{N,22} \gg c_{N,12}$.** In this case, the signal is dominated by $a^* \to N_1 N_1$ processes since the larger centre-of-mass energy required for $a^* \to N_2 N_2$ leads to PDF-induced suppression, and the analysis therefore follows Sect. 3. This case was already discussed in details for the gluon-philic ALP scenario in Ref. [160], and was extended here to all gauge couplings. Remarkably, the presence of EW couplings requires the extension of the JALZ signals with two extra jets, which can help disentangle the effects and impact of such couplings.

**Dominant off-diagonal interactions, $c_{N,12} \gg c_{N,11}, c_{N,22}$.** In this case the analysis follows Sect. 4. The signal is overall similar to the one studied in the previous case, but the cascade emission of an additional ALP could lead to extra photons, fermions, jets, or missing energy in the final state, depending on the ALP lifetime. Such a signal would not only give us information on the flavour structure of HNLs, but would also be a smoking gun for the presence of an ALP as mediator. If a large HNL mass splitting is considered, the PDF-suppression may compensate the $c_{N,12}$ dominance and the analysis should include both the topologies with and without cascades.

**Dominant $c_{N,22}$.** In this case, the specific decay channel depends on the relative value of $c_{N,12}M_{N_2}/f_a$ compared to the HNL mixing parameters $\Theta$. If $c_{N,12}/f_a \ll \Theta/M_{N_2}$, the decays mediated by the HNL mixing dominate and each $N_2$ undergoes a prompt semileptonic decay, including $N_2 \to W\ell$. This case amounts to the JALZ signal discussed in Sect. 3. If on the other hand, $c_{N,12}/f_a \gg \Theta/M_{N_2}$, the new ALP-HNL interaction dominates, and each $N_2$ separately undergoes a cascade decay into $N_1$ and an on-shell ALP: $a^* \to N_2N_2 \to (N_1a)(N_1a)$. The phenomenology associated with each ALP is then the same as in the case of $c_{N,12}$ domination, but the presence of two of them will lead to a higher amount of missing energy or a busier final state, neither of which affects the analysis method proposed in Fig. 7.

## 5    Discussion and conclusions

In this work, we have analysed new signals that arise when an ALP interacts with HNLs. Since the coupling of fermions to the ALP is proportional to their mass, its interaction strength with a hypothetical TeV-scale HNL would be greatly enhanced compared to all other ALP-SM interactions. This feature can be exploited to test HNL physics in a way otherwise impossible. As previously detailed in Ref. [160], a direct ALP-HNL coupling makes the production of two on-shell HNLs simpler than the conventional mono-HNL production via SM interactions: on the one hand, it eliminates the $\Theta$ mixing suppression from the amplitude, which makes HNLs production via SM interactions virtually impossible in the traditional Type-I Seesaw; on the other hand, the simultaneous presence of two HNLs greatly reduces the SM background, thus considerably enhancing the sensitivity of a direct search, as detailed in Sect. 3.3. Additionally, different mediators for the production of the HNL pair have been considered, with the $Z'$ offering the closest phenomenology. However, several observables would be sensitive to this spin-1 mediator. Due to the scalar nature of the ALP, the final state HNLs are necessarily produced with opposite spins. If this requirement is not satisfied, the mediator must have a non-zero spin, such as the $Z'$. At the muon collider, s-channel production of HNLs via a $Z'$ mediator is expected, a process that is not possible with an ALP and leads to a different experimental signature.

In Sect. 3, we first considered a generic ALP Lagrangian coupled to a single HNL via the derivative interaction $c_N \partial_\mu a \overline{N_R}\gamma^\mu N_R$, and we derived the sensitivity reach to the ALP couplings at the High-Luminosity LHC and at a proposed muon collider, showing that strong sensitivity can be achieved for various benchmark models.

In Sect. 4, we then considered the case in which two HNLs $N_{1,2}$ are present. This is a somewhat more realistic scenario if one aims to explain neutrino masses via a Seesaw mechanism, as detailed in Sect. 2.1. In this case, $c_N$ is a matrix, possibly with off-diagonal entries coupling the ALP to $N_1N_2$. Such a scenario opens up new processes and signatures, the most prominent one being the possibility of having the heaviest HNL decay into the lightest one while emitting an ALP.

Both scenarios are complementary since, as detailed in Sect. 4.4, they allow probing different textures of the $c_N$ matrix: while we generically expect PDF-induced suppression to favour the production of the lightest HNL $N_1$, a $c_N$ that is hierarchical or contains a strong off-diagonal component could compensate for this suppression and lead to a variety of processes featuring cascade HNL decays. Furthermore, the distinct experimental signatures in these scenarios could help disentangle the contributions of the various elements of $c_N$.

In conclusion, the exploration of ALP-HNL interactions not only opens new avenues for probing HNL physics beyond conventional methods, but also significantly enhances our potential to uncover the subtle dynamics of ALPs and their role in the broader landscape of particle physics.

## Acknowledgments

The authors thank José Miguel No for useful discussions during the first stage of the project.

**Funding information**   The authors acknowledge partial financial support by the Spanish Research Agency (Agencia Estatal de Investigación) through the grant IFT Centro de Excelencia Severo Ochoa No CEX2020-001007-S and the grant PID2022-137127NB-I00 funded by MCIN/AEI/10.13039/501100011033/ FEDER, UE, and by the European Union's Horizon 2020 research and innovation programme under the Marie Skłodowska-Curie grant agreement No 860881-HIDDeN and 101086085-ASYMMETRY. JLT acknowledges partial support from the grant Juan de la Cierva FJC2021-047666-I funded by MCIN/AEI/10.13039/ 501100011033 and by the European Union "NextGenerationEU"/PRTR. This article is based upon work from COST Action COSMIC WISPers CA21106, supported by COST (European Cooperation in Science and Technology).

## A   Cross section computation using MadGraph

The cross sections and lifetimes reported throughout this paper have been computed using `MadGraph5_aMC@NLO` (v3.5.3) [217], and cross-checked against various analytical estimates. In order to simulate the processes described in Sect. 3 and 4, we have implemented the Lagrangian from Sect. 2 in the `Mathematica` (v12.3.1) [218] package `FeynRules` (v2.3.49) [219], taking some inspiration from the existing models `HeavyN` [220, 221] and `ALP_linear` [143]. The model is then exported to the UFO format [222, 223] which can be loaded into MadGraph to extend its generation capabilities to the relevant BSM processes.

The generation is performed at leading order, with the parton shower and hadronisation simulated by `PYTHIA` (v8.306) [224, 225] for the production at the LHC, using the MLM merging algorithm with a fixed $k_T$ scale of 75 GeV (selected by requiring well-behaved matching plots). The merging procedure was found to only affect the cross sections of the gluon-philic benchmark at the LHC since it is the only scenario that features a variable number of (possibly soft or collinear) jets in the final state. In all other scenarios, the cross sections can be well-approximated by the parton-level cross sections. Since the LHC is a hadron collider, the corresponding cross sections crucially depend on the PDFs used. Here, we have used the PDF set `NNPDF40_nnlo_as_01180` (`lhaid = 331100`) from the NNPDF collaboration [226] through the `LHAPDF6` interface [227], evaluated using the default choice of scale (`dynamical_scale_choice = -1`) in MadGraph. We only consider production involving the four lightest quark flavours $u, d, s, c$ and their charge-conjugates, the contribution from heavier quarks having been estimated to be small. When applicable, we use as numerical inputs $\alpha_{\rm EW} \approx 1/128$, $\sin^2(\theta_W) \approx 0.231$ [228] and we use `CRunDec` [229] to evaluate $\alpha_S$ at a scale equal to the sum of the masses of the two produced HNLs. No generation cuts have been applied in either experiment, beyond the default pseudorapidity cuts of $|\eta| < 5$ for jets and $|\eta| < 2.5$ for charged leptons and photons.

In order to keep under control the combinatorics resulting from flavour, the production processes for the HNLs and their decays are simulated separately, using the narrow-width approximation to obtain the overall cross-section of each process. We first evaluate the total production cross sections for all five benchmarks listed in Tab. 1 at two different mass points (400 GeV and 1600 GeV), as already discussed in Sect. 3.1, in order to study which production mechanisms are dominant. For the two retained benchmarks (gluon-philic and electroweak-philic), we then simulate separately each of the processes listed in Fig. 4, scanning over the HNL mass between 130 GeV and ∼ 5 TeV. We then perform two additional scans: one for the

total width of the lightest HNL $N_1$ and one for its partial width into the fully-reconstructible channel $N_1 \to W\ell \to \ell jj$, allowing us to derive the branching ratio of the latter. For the heaviest HNL $N_2$, we compute its decay width into $N_2 \to N_1 a$, which dominates its total width for sufficiently small mixing angles $\Theta$.

Throughout the above calculations, the coefficients $c_X/f_a$, mixing angles $\Theta$ and decay widths are set to small, fixed values. The physical widths and cross sections can then be obtained using the rescaling method described in Ref. [160], which relies on the narrow-width approximation and the scaling properties of the signal:

$$\sigma \propto \frac{c_X^{\text{eff}\,2} c_{N,IJ}^2}{f_a^4} \frac{|\Theta_{\alpha I}|^2 |\Theta_{\beta J}|^2}{\Gamma_{N_I} \Gamma_{N_J}} \propto \frac{c_X^{\text{eff}\,2} c_{N,IJ}^2}{f_a^4} \frac{|\Theta_{\alpha I}|^2}{|\Theta_I|^2} \frac{|\Theta_{\beta J}|^2}{|\Theta_J|^2} , \tag{A.1}$$

where $c_X^{\text{eff}}$ and $c_{N,IJ}$ denote the Wilson coefficients involved in the considered production process, $f_a$ is the ALP scale, $\Gamma_{N_I}$ is the width of the HNL $N_I$, $\Theta_{\alpha I}$ is the mixing angle between $N_I$ and $\nu_\alpha$, and we have defined $|\Theta_I|^2 = \sum_{\alpha=e,\mu,\tau} |\Theta_{\alpha I}|^2$. After summing over the (rescaled) production channels (except those with opposite-sign same-flavour lepton pairs), the total cross section $\sigma_{\text{total}}$ can then be used to derive the expected exclusion limits at 95% CL (if no signal is observed, in the almost-zero-background regime) by requiring $\epsilon_{\text{sig}} \mathcal{L}_{\text{int}} \sigma_{\text{total}} = 3$, with $\mathcal{L}_{\text{int}}$ the integrated luminosity at the considered experiment and $\epsilon_{\text{sig}}$ the overall signal efficiency. Computing the latter is beyond the scope of this work since it is expected to greatly depend on the treatment of jets, but an estimation for the non-cascade case was performed in Ref. [160, App. B].

Like any sensitivity estimate, our results are approximate and subject to a number of uncertainties, even within our stated assumptions (e.g. ignoring analysis cuts and the background). The statistical uncertainty coming from the Monte-Carlo event generation, as well as the systematic uncertainties associated with the narrow-width approximation and the overall analysis code (estimated by trying to reproduce the numbers from Ref. [160]), were all estimated to be at the few-percent level or below. While ignoring the $\mathcal{O}(1)$ loop prefactors in Tab. 1 obviously introduces $\mathcal{O}(1)$ errors for the benchmarks in which loop-induced couplings dominate, they do not affect the two benchmarks (gluon-philic and electroweak-philic) for which we have reported the limits, and our limits can be easily recast for the "BM" benchmarks from Tab. 1 by performing a suitable rescaling (that can include the precise $\mathcal{O}(1)$ loop prefactors if desired). The main irreducible source of uncertainty for this analysis therefore comes from the PDFs and the scheme used to determine the scale at which they are evaluated. Since the associated uncertainty strongly depends on the HNL mass (through the partonic centre-of-mass), reporting a single number can be difficult. However, replacing the PDF set with an older version of NNPDF lead to a change in cross-section of only a few percent towards low HNL masses, while the heaviest HNLs saw $\mathcal{O}(1)$ relative changes (but lower absolute changes, since their production is kinematically suppressed). Overall, any discrepancy between the present sensitivity estimate and a potential future search is more likely to come not from these uncertainties, but from analysis cuts, detector and reconstruction efficiencies, and the background.

## B  Background estimation

In Sects. 3 and 4, we have derived the couplings $(c_N c_{\widetilde{X}})^{1/2}/f_a$ necessary to produce 3 events with luminosity $\mathcal{L} = 3\,\text{ab}^{-1}$. In the absence of background, this is essentially equivalent to 95% C.L. In this section, we attempt to estimate the order of magnitude of the background effect. To do so, we rescale the single HNL background estimated in Ref. [18] in the same

sign-$e\mu$ channel

$$N_{bkg}^{\text{2HNLs}}(\mathcal{L}=3\text{ab}^{-1}) = 2 \times \frac{3 \times 10^3}{35.8} N_{bkg}^{\text{1HNL}}(\mathcal{L}=35.8\text{fb}^{-1}), \tag{B.1}$$

where the factor 2 takes into account the double HNL production.

We focus now on the number of events necessary to set constraints at 95% C.L., $N_{95\%}$. To calculate this we employ Poisson's Cumulative Distribution Function (CDF), which describes the probability of observing a number of events $x$ smaller or equal to a certain value $k$ when these events occur randomly following a Possion's distribution with mean value $\lambda$. The CDF is then given by

$$F(k|\lambda) \equiv P(x \le k|\lambda) = \sum_{n=0}^{k} \frac{\lambda^n}{n!} e^{-\lambda}. \tag{B.2}$$

In our case, the mean rate is the background itself, $N_{bkg}$, and we want to determine when the needed observed events $N_{95\%}$ is larger than the background expectation; therefore

$$F(N_{95\%} - 1|N_{bkg}) \ge 95\%, \qquad N_{95\%} = F^{-1}(0.95|N_{bkg}). \tag{B.3}$$

In our case, $N_{\text{signal}}^{e\mu}$ at $\mathcal{L}=3\text{ab}^{-1}$ are the values we obtained in Fig. 6 divided by a factor 9 since we only compare the same sign-$e\mu$ signal-background. The results for different values of $M_N$ can be seen in Tab. 5.

The ratio between the two estimated bounds without and with the background can be seen in Fig. 13. As can be seen, for small HNL masses the difference can be as large as a factor of $\sim 10$, while it rapidly reduces to $\sim 6$ for $M_N \gtrsim 500$ GeV. Finally, recall that such an estimation is quite pessimistic. It naively doubles the background of the single HNL searches instead of employing the search strategy to use the presence double invariant mass strategy. This may substantially change the situation and bring the results closer to the no-background hypothesis.

Table 5: Estimated background by employing single HNL results from Ref. [18] compared to the number of necessary events to set 95% C.L. to the signal assuming $(c_N c_{\widetilde{X}})^{1/2}/f_a = 1$ TeV$^{-1}$.

| $M_N$ [GeV] | $N_{bkg}^{\text{1HNL}}(\mathcal{L}=35.8\text{fb}^{-1})$ | $N_{bkg}^{\text{2HNLs}}(\mathcal{L}=3\text{ab}^{-1})$ | $N_{95\%}$ | $N_{\text{signal}}^{e\mu}$ ($\mathcal{L}=3\text{ab}^{-1}$) |
|---|---|---|---|---|
| 200 | 11.1 | 1855.2 | 1928 | $2.1 \times 10^5$ |
| 300 | 5.8 | 969.4 | 1023 | $1.1 \times 10^5$ |
| 400 | 2.2 | 367.7 | 401 | $7.3 \times 10^4$ |
| 500 | 1.8 | 300.8 | 331 | $4.9 \times 10^4$ |
| 600 | 1.2 | 200.6 | 226 | $3.0 \times 10^4$ |
| 800 | 1.6 | 267.4 | 296 | $1.4 \times 10^4$ |
| 1000 | 1 | 167.1 | 191 | $6.8 \times 10^3$ |
| 1200 | 1 | 167.1 | 191 | $2.9 \times 10^3$ |
| 1500 | 0.8 | 133.7 | 154 | $1.1 \times 10^3$ |
| 1700 | 0.8 | 133.7 | 154 | 460 |

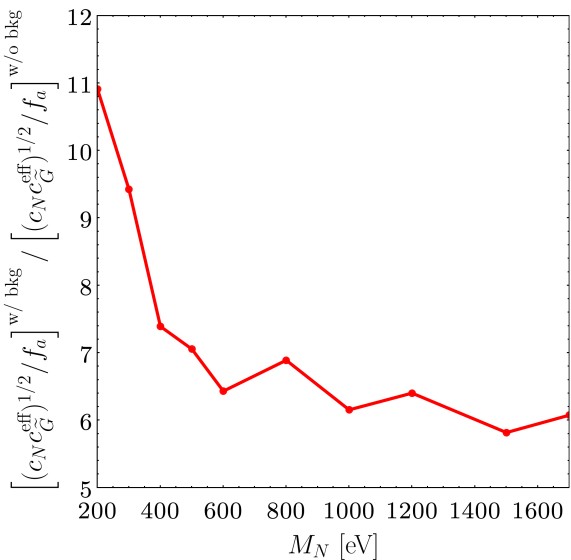

Figure 13: Ratio of the constraints $(c_N c_{\widetilde{X}})^{1/2}/f_a$ estimated with (w/) and without (w/o) background using the results from Tab. 5 as a function of the HNL mass.

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
