# Peer review of "ALPs and HNLs at LHC and Muon Colliders: Uncovering New Couplings and Signals"

_SciPost Physics, doi:SciPost Phys. 18, 084 (2025)_

## Round 1 · Referee Report · Anonymous (Referee 1) · 2024-9-13

Strengths
Comprehensiveness
Supplementary details, extensive bibliography
Weaknesses
Discussion of experimental backgrounds and complementary ALP searches
Report
The article is quite lengthy with many detailed explanations and an extensive bibliography. Given that the goal is to fill in some additional details to a previous study of this model, this is not unwelcome. Nevertheless, there is a risk that the reader loses focus. It might help to state the goal and the main findings of the study more clearly in the introduction and to introduce the main processes and signatures at an earlier point in the paper, so that the reader can distinguish relevant from supplementary information.
The analysis itself is detailed and thoughtful and the results are interesting. I could imagine that these kinds of models and signatures will be an attractive target for future searches at the LHC and beyond. The paper therefore can potentially be published in SciPost. However, the following points need to be addressed first:
Requested changes
1) The authors seem to take a lot of the motivation for their study from the fact that assuming comparable Wilson coefficients for different fermions in the EFT approach, ALPs will couple dominantly to the heaviest fermions. However, for this argument to be convincing, the authors should demonstrate that indeed similar Wilson coefficients are generated for the different fermions in plausible UV completions. For SM fermions this can be motivated through the assumption of minimal flavour violation. But how do the HNLs feature in this argument?
2) I am not convinced by the way how the authors treat the loop-induced couplings to SM gauge bosons for the case of ALPs coupled to SM fermions. These couplings arise through one-loop diagrams, which have a complicated dependence not only on the centre-of-mass energy, but also on additional mass scales, such as the mass of the fermion in the loop or the momentum transfer to additional gauge bosons radiated from the virtual fermions. It is not possible (probably not even for an order-of-magnitude estimate) to estimate these effects using RGE-induced effective couplings to gauge bosons, which have no dependence on the kinematics. The authors need to clarify the approximate nature of their treatment and quantify the resulting uncertainty.
3) I find the claim that the signature is without background at $3\,\mathrm{ab}^{-1}$ very bold. This seems plausible for the case of same-sign leptons, but for the opposite-sign leptons, there will be a smooth SM background, on top of which a peak has to be identified. I would encourage the authors to demonstrate more explicitly, that the SM background is small in the kinematic regions of interest.
4) The authors discuss that decays of on-shell ALPs are not strongly constrained for ALP masses between 0.1 GeV and 10 GeV. This is broadly correct right now, but will it remain true until 2040? In other words, is there really no chance to directly detect on-shell ALPs in this mass range on similar timescales? Presumably both the HL-LHC and a muon collider should have some sensitivity, as will other experiments. Is it really easier to search for the ALP in the decay of HNLs compared to direct production from SM particles?
5) Is it actually possible (using for example angular correlations) to distinguish the case of HNL production via ALPs from other scenarios? Or can the ALP only be identified in the scenario with two HNLs?
6) The numbers in table 2 require a choice of $c_N$, which is however not stated prominently in the text.
7) What does JALZ stand for? If it's just a different way of writing $j4\ell2$, I would much prefer the latter for clarity.
Recommendation
Ask for major revision
We answer the questions/comments made by the referee:
There are many models which could generate an ALP EFT with O(1) Wilson coefficients for SM fermions and HNLs. In fact, in any UV ALP model such coefficients are typically associated with the PQ charges of the fermion fields, which are $\mathcal{O}(1)$ factors. Furthermore, the proportionality to the fermion masses is inherited by the derivative structure of the ALP couplings to fermions. We added a discussion about the topic with some examples in the main text.
The 1-loop contribution of the ALP to gauge bosons has indeed a rich momentum structure and depends on the fermion masses. However, in the high momentum transfer limit, $p^2\gg v^2$ (which is our case as the HNLs are much heavier than the EW scale), the momentum dependence (alongside the fermion masses) drops, as EW symmetry gets restored and the relations we write become valid up to $\mathcal{O}(1)$ colour/hypercharge factors which do not change nor qualitatively nor quantitatively any of the main conclusions. We modified the text to make this point clearer and alongside references where such claims can be verified.
We have clarified that only processes with same-sign or different-flavour charged leptons have no SM background. In the case of opposite-sign, same-flavour leptons, we mention that some SM background is expected, but that the unusual kinematic structure of the process should help reduce it (like other, non-SM background components, such as combinatorial background). Estimating the SM background for opposite-sign, same-flavour events would likely require a dedicated analysis, since it is very sensitive to the tails of the double invariant-mass distribution shown in Fig.~7, which cannot be easily simulated with MadGraph due to the very high statistics required.
The referee is correct in saying that future experiments could hopefully be sensitive. However, this type of analysis strongly depends on the details of the experiments. We have already considered the possible outcome of HL-LHC and muon colliders.
Different mediators for the production of the HNL pair have been considered in the literature and the $Z′$ case offers the closest phenomenology. Due to the scalar nature of the ALP, the final state HNLs are necessarily produced with opposite spins. If this requirement is not satisfied, the mediator must have a non-zero spin, such as the $Z’$. We add a comment about this in section 5.
We specified the assumed value.
We clarified this on page 3 and we kept the JALZ notation to be consistent with our previous publication.
I am satisfied with most of the replies to my previous report and the corresponding changes to the manuscript. The only remaining issue, as pointed out by the other referee, are the background estimates. I tend to agree that one cannot simply assume the absence of backgrounds at 3 ab$^{-1}$ without further calculation, even for relatively exotic signals. Even if there are no Standard Model backgrounds, the issue of misidentification or misreconstruction remains, in particular in the upcoming environment of very high pile-up. Having said that, I think it is clear that such a study is far beyond the scope of the present work, which is already quite extensive. Moreover, the precise position of the sensitivity curves is largely irrelevant for the main point of the paper, so that even a certain degradation of sensitivity would not alter the main conclusions. My suggestion is therefore to rephrase some of the text in a more careful way. For example, instead of a projected sensitivity at 95% CL, the authors could refer to their curves as lines of 3 predicted events, which would correspond to the sensitivity in the absence of background. Instead of showing lines for 300 fb$^{-1}$ and 3 ab$^{-1}$ one would then simply show 3 and 30 predicted events at 3 ab$^{-1}$, such that the actual sensitivity can be read off once the background level is known.
We answer the questions/comments made by the referee:
We thank the referee for the reply to our answers and for his opinion that ``it is clear that such a study is far beyond the scope of the present work, which is already quite extensive'' referring to a more elaborate estimation of the background. We also thank her/him for the suggestions on how to rephrase part of the text.
We implemented all the suggestions accordingly.
We also added in App. B a conservative numerical estimation of the background using the reference proposed by the other referee. This should be enough to get an intuition of the background effect.
The modifications can be seen in red in the new version.

Author: Arturo de Giorgi on 2024-11-26 [id 4993]
(in reply to Report 2 on 2024-09-19)We reply below to the comments/questions made by the referee:
The referee is correct in her/his comment as we were referring in our sentence to collider possibilities, overlooking the other searches. To be correct, we modified the sentence, softening the statement.
We prefer to leave the references for completeness and fairness.
We changed the sentence to stress out the different possible outcomes. In the case of same-sign or different-flavour final leptons, there is no SM background. On the other side, in the case of different-sign and same-flavour final leptons, there is indeed a SM background, whose estimation requires a dedicated analysis, as we pointed out in the present version of the paper. However, in our setup, the theoretical prediction is greatly insensitive to the flavour of the final leptons, due to the democratic structure of the mixing matrix. This is also the case for the sign of the final leptons, as typically occurs with on-shell HNLs.
[4,5] We modified the discussion in page 17 and the following pages. Since the SM background comes from the tails of the distribution, its numerical estimation requires generating samples with very large statistics, which is complicated both from a numerical and computational perspective. On the other side, applying a cut on our possible signals in order to remove same-flavour opposite-sign final state leptons, we end up with $\approx 5/6$ of the number of expected events with no SM background. We believe that in this way the presentation of the results is much cleaner and at the same time does not affect the qualitative results, only slightly modifying the quantitative ones. We changed Fig.~6 and the following figures that depend on it and the text on page 17 and following ones.
As we commented on page 17, the dependence on the efficiency is with the quartic root and therefore the plot is very mildly affected by an efficiency different from 1. We modified the plot labels to include the efficiency to be more general. On the other hand, we are also restricting processes without an SM background.
In fig.12 we assumed $100\%$ efficiency and we explicitly specified it in the caption.
The masses of the ALPs under consideration are chosen for their phenomenological significance. Heavier or lighter masses produce nearly identical phenomenological signatures. For instance, as shown in Table 4, ALP masses either below or above the chosen range result in identical final states in their decay. In any case, to maintain the validity of the Lagrangian, the ALP mass must remain smaller than the scale $f_a$.
Anonymous on 2024-11-29 [id 5007]
(in reply to Arturo de Giorgi on 2024-11-26 [id 4993])Dear authors, Dear Editor
I want to thank the authors for their response, which addressed some of the comments of the previous report.
However, the authors do not address my main concerns, regarding the background and signal identification, in a satisfactory way.
In the original version, the authors claimed that the signal signature (with a 2l+4j final state) is fully reconstructible and does not have SM backgrounds. In their reply, the authors admit that for the different-sign same-flavour di-lepton channel there is indeed a SM background (although their size is not estimated). The authors, however, still claim that there are no SM backgrounds in the case of same-sign or different-flavour final state leptons. No quantitative estimate or references is provided to back up this claim, which is crucial for the main results presented in the paper.
An opposite-sign different-flavor final state can easily be produced, for example by fully leptonic top-quark pair decays. The corresponding cross section is about 100 pb, and therefore orders of magnitude larger than the considered signal. A careful background analysis is needed to estimate these backgrounds.
Backgrounds for a same-sign di-lepton final state are indeed smaller, but still produced by a variety of SM processes. This includes di-boson production such as W+W+, WZ or Z; vector boson associated top production such as ttW or ttZ; and heavy flavor associated production, for example Zb/Zc, where the heavy meson decays leptonically, just to name a few. In addition, there are experimental backgrounds from particle misidentification or wrong charge identification. These scenarios have been discussed extensively in the literature, including both phenomenological studies and experimental searches, for example in the context of super-symmetric particle searches (see 1605.03171) and also HNL searches (1806.10905). Note that this list of references is by no means complete, and I strongly encourage the authors to study the existing literature more extensively. For example, 1806.10905 shows that thousands of background events for a same-sign di-lepton signature (with lepton being a muon or electron) have been measured for a luminosity of 36 fb^-1. Therefore hundred thousands of such events are expected during the HL-LHC era with 3000 fb^-1. Therefore, simply claiming that there are no SM backgrounds is unfortunately not correct.
The arguments and consideration presented above need to be presented in the paper.
The number of background events can likely be reduced using kinematic cuts, and a dedicated simulation study is needed to quantitative these backgrounds. A standard chain of tools exist for this purpose. At the same time, the application of cuts cuts will reduce the signal efficiency. For example, in 1806.10905 the combined acceptance and efficiency was around the percent level. When presenting results, a more realistic estimate for both backgrounds and signal efficiency need to be considered.
I again ask the authors to include a quantitative estimate of backgrounds and signal efficiency. Without such a study, I cannot assess the correctness of the presented results and therefore do not consider the paper suitable for publication in a scientific journal.
Anonymous on 2025-01-24 [id 5150]
(in reply to Anonymous Comment on 2024-11-29 [id 5007])We thank the referee for the detailed answer and for bringing to our knowledge the work 1806.10905 (Ref. [18] of the present version of our manuscript) which greatly helped to perform the numerical estimation the referee was asking for.
In particular, we created App.B with the details of the numerical estimation of the background obtained from the results of Ref. [18]: we adapted their result considering that our signals involve 2 on-shell HNLs.
We obtained the number of background events as a function of the HNL mass for the gluon-philic case and compared with the expected number of events of our process. This can be read out in Tab.5.
Moreover, we translated this information in terms of the effect of the bounds on the Lagrangian parameters, as shown in Fig.~13. We find that the background can weaken the background-free constraints by ~6.
Last to say that this is just a very pessimist estimation as we just doubled the background from Ref. [18] instead of computing explicitly the background and implementing a tagging strategy for our process. Doing so can be properly done from the experimental collaboration, the ratio in Fig. 13 can only improve.
We adapted the main text in several parts, making explicit mention of the results of App.B and having the wording about the background consistent with these results. Now the main text should be free of any ambiguity.
We understand that this numerical estimation was the last request from the referee and therefore hope that the paper will be accepted in its present form.

---

## Round 1 · Referee Report · Anonymous (Referee 2) · 2024-9-19

Strengths
- well written
- clear presentation
- extensive theory section
Weaknesses
- lack of background estimates
- lack of efficiency estimate
Report
The authors then first discuss the phenomenology of a single HNL, focussing on the pp > a* > NN > 2l2W channel, which is referred to as JALZ topology. The authors then proceed to the two HNL case, focussing on the pp > a* > N1 N2 > N1 N1 a > 2l2W a channel. For both cases, they estimate the rate for this process and (assuming perfect signal efficiency and negligible backgrounds) obtain projections on the model parameters, and conclude that "strong sensitivity can be achieved for various benchmark models".
The manuscript is well organized, the presentation is clear and the work seems technically correct.
My main concern regards the underlying assumptions of perfect signal efficiency and negligible backgrounds when estimating the projected sensitivities. In particular, there may be sizable combinatorial background or backgrounds where light jets were mis-tagged as hadronic W. Without a quantitative estimate, at least at the order of magnitude level, it is not clear whether the made assumptions are valid, and whether the main conclusion holds. Therefore, I cannot recommend publication at the moment. A realistic estimate of background rates and efficiencies need to be added before the paper can be considered for publication.
Requested changes
1) You say that "these effects cannot be observed experimentally". Neutrinoless double beta decay experiments are in principle able to experimentally observe effects of large Majorana masses.
2) The last 3 paragraphs of page 2 have 150 references. That seems a bit excessive. Please reconsider if all references are truly needed.
3) "the process has a fully reconstructible final state with no SM background": If I see correctly, you have a 2l+jets final state. This could be faked at the LHC, for example by tt+jets. Although the various on-shell conditions will clearly help to reduce such backgrounds, claiming 'no SM background' is likely misleading.
4) Does Fig 6 include any selection cuts (pt, eta, deltaR) and efficiencies? If yes, the assumption should be described in the text. If not, this should be clearly stated in the caption.
5) Page 17 qualitative explains why we would expect backgrounds to be low and how such a search could be performed. However, it does not attempt to estimate these backgrounds. It is therefore not clear if the assumption of negligible combinatorial backgrounds is correct. In particular, note that there is a sizable rate of quarks/gluons being misidentified as hadronic W-bosons. Looking at ATL-PHYS-PUB-2023-020, for a signal efficiency of 50% the misidentification rate is still about 1%. Given the large rate of jets at the LHC, this could cause a substantial background. I strongly suggest obtaining some quantitative background estimate to validate this assumption using established tool chains, for example using Madgraph / Pythia / Delphes.
5) In Fig 8 you show the expected sensitivity. Do I understand correctly that this is obtained directly from Fig 6 by requiring at least 3 events or so, so using the explicit assumption of vanishing backgrounds and 100% signal efficiency. If yes, that should be clearly stated in the caption/text. Since you didn't actually show that these backgrounds are indeed negligible, it is not clear if those results are actually correct.
6) Fig 12 shows sensitivity results. Do I understand correctly that these again assume negligible backgrounds and 100% signal efficiency? As before, estimating the sensitivity requires a background estimate and should include efficiencies.
7) The result figures consider a very light ALP, around GeV masses. Why was this chosen, in comparison to, for example, weak scale mass ALPs? Please clarify.
Recommendation
Ask for major revision

---

## Round 3 · Author Response

We implemented all the referees' recommendations.

---

## Round 3 · List of Changes

The modifications are shown in red in the article.

---

## Editorial Decision

published